**Title page**

5  Hypoxia occurs also in small highly turbid estuaries: the example of the Charente (Bay of Biscay)

Sabine Schmidt[1], Ibrahima Iris Diallo[2]

[1] Univ. Bordeaux, CNRS, Bordeaux INP, EPOC, UMR 5805, F-33600 Pessac, France

[2] Rectorat de Bordeaux, Bordeaux, 33000, France

*Correspondence to*: Sabine Schmidt (sabine.schmidt@cnrs.fr)

# Hypoxia also occurs in small highly turbid estuaries: the example of the Charente (Bay of Biscay)

Sabine Schmidt[1], Ibrahima Iris Diallo[2]

[1] Univ. Bordeaux, CNRS, Bordeaux INP, EPOC, UMR 5805, F-33600 Pessac, France
[2] Rectorat de Bordeaux, Bordeaux, 33000, France

*Correspondence to*: Sabine Schmidt (sabine.schmidt@cnrs.fr)

Key works: estuary, Charente, turbidity maximum, dissolved oxygen, monitoring, high-frequency, long-term trend

**Abstract.** The French coast facing the Bay of Biscay (north-east Atlantic) is characterised by the presence of small macrotidal and turbid estuaries, including the Charente, which is geographically located between the two large estuaries of the Gironde and the Loire (south-west France). Multi-year, multi-site, high-frequency water quality surveys have shown that the Loire, and to a lesser extent the Gironde, suffer from summer hypoxia. These observations raised the question of the possible occurrence of hypoxia, particularly in one of these small estuaries, the Charente, which flows into the Bay of Marennes-Oléron, the first oyster farming area in France. Unlike its two large neighbours, the Charente estuary is not continuously monitored, although it is subject to similar climatic changes and anthropogenic pressures, making it impossible to assess potential risks to the ecosystem. Here we present a first study of dissolved oxygen in the Charente estuary, based on a combination of longitudinal studies along the estuary axis and instrumented sites to determine the intensity and spatial extent of deoxygenation. Temperature, dissolved oxygen and conductivity/salinity sensors were deployed at several sites during the summers of 2018, 2019 and 2020 to record temperature, salinity and dissolved oxygen every 15 minutes. The high-frequency dataset is compared with a long-term low-frequency dataset (1975-2022; 8-12 measurements per year) to determine whether or not there is a deterioration in the oxygenation of the Charente estuary. The high-frequency dataset shows a high variability of dissolved oxygen (DO) with a clear influence of the tidal cycle. During summer, DO concentrations are often below 5 mg L$^{-1}$ and sometimes even below 2 mg L$^{-1}$, indicating the presence of a summer estuarine oxygen minimum zone with an extension along the estuarine axis of about 20-25 km. Temperature is the main factor controlling DO in the Charente estuary, which limits preventive management strategies and, in the context of global warming, raises questions about the long-term suitability of estuarine conditions for the needs of the biological communities, particularly migratory fish.

## 1 Introduction

Globally, there has been a decrease in dissolved oxygen (DO) in the oceans, which is more pronounced in coastal waters, resulting in more frequent hypoxia events characterised by DO concentrations below 2 mg L$^{-1}$ (Diaz and Rosenberg, 2008; Zhang et al., 2010; Stramma et al., 2011; Gilbert et al., 2010). Dissolved oxygen is an important parameter for diagnosing water quality. Essential for the survival of aquatic species, oxygen levels are the main factor determining the type and abundance of organisms that can live there (Breitburg, 2002; Sampaio et al., 2021). Dissolved oxygen comes from two natural processes: diffusion from the atmosphere and photosynthesis by aquatic photosynthetic organisms. The mixing of surface waters by wind and waves increases the rate at which oxygen from the air can be dissolved into the water. Conversely, bacteria and other decomposers use oxygen to break down organic matter. It is therefore common to observe deviations from the theoretical saturation value predicted by temperature and salinity (Benson and Krause, 1984). When large inputs of nutrients, usually from wastewater discharges, stimulate excessive growth of aquatic photosynthetic organisms (eutrophication), their decomposition can lead to oxygen depletion, even to hypoxia (< 2 mg L$^{-1}$) or anoxia (0 mg L$^{-1}$). Stratification of the water column can also lead to oxygen depletion in bottom waters. Conversely, dissolved oxygen can become oversaturated due to photosynthesis, usually during the day. Given the diversity of processes affecting DO, the variability of dissolved oxygen in

coastal systems is still poorly documented, making it difficult to estimate its evolution over several decades. In particular, the prediction of DO trajectories in estuaries is more complex than in coastal waters due to hydroclimatic but also anthropogenic

forcing (Villate et al., 2013). Indeed, during the 20th century, many estuaries adjacent to cities experienced significant water quality degradation due to excessive discharges of untreated domestic and industrial wastewater. Severe hypoxia was observed in estuaries such as the lower Hudson, Thames and Scheldt, among others (Brosnan and O'Shea, 1996; Billen et al., 2005). However, during the 1980s and 1990s, improvements in wastewater collection and treatment helped to improve DO levels in some estuaries (Andrews and Rickard, 1980; García-Barcina et al., 2006).

Situated at the interface between the ocean and the continent, estuaries are dynamic natural collectors of liquid and solid inputs from watersheds that are highly variable both spatially and seasonally. Downstream, the large volumes of seawater entering with each rising tide are progressively diluted, affecting the chemistry of estuarine waters. The physico-chemical quality of estuarine waters determines the distribution of species on which many ecosystem services (e.g. commercial or recreational fisheries) depend. Estuaries are habitats of great ecological interest for many biological groups, including fish, which use them

as nurseries or spawning grounds (Delage et al., 2019). In macrotidal estuaries with a tidal range greater than 3 m, tidal asymmetry causes longer and weaker ebb currents and shorter and stronger flood currents. This results in net upstream transport and trapping of suspended sediment in the inner and upper reaches, leading to the formation of a mobile region with high suspended particulate matter (SPM) concentration known as the turbidity maximum zone (TMZ) (Allen et al., 1980). The TMZ is characterised by SPM loads of several hundred milligrams to several grams of dry sediment per litre (Uncles et al., 2002).

Such high particle concentrations affect water chemistry by limiting gas exchange with the atmosphere (Abril et al., 2009). In addition, high particle concentrations limit light penetration, reducingphytoplankton primary production. However, they also provide substrate for microbial activity, promoting mineralisation of organic matter and DO consumption (Goosen et al., 1999). These factors are likely to contribute to deoxygenation in macrotidal turbid estuaries (Talke et al., 2009).

Along the French coast facing the Bay of Biscay (Northeast Atlantic), there are many small tidal and turbid estuaries,

geographically located between the two large estuaries of the Gironde and the Loire, which are natural corridors for migratory fish (Arevalo et al., 2023). The spatio-temporal DO dynamics of the two large systems are well described by long-term, high-frequency and multi-site water quality monitoring. While episodic summer hypoxia events have been recorded in the fluvial Gironde estuary, the Loire estuary experiences permanent summer hypoxia in its lower reaches (Lanoux et al., 2013; Schmidt et al., 2019). These observations have raised the question of the possible occurrence of hypoxia in the small estuaries between

the Loire and the Gironde. Indeed, the question of the evolution of the quality and ecological status of estuarine waters will become increasingly critical in the face of rising temperatures and sea levels (Simon et al., 2023; Chust et al., 2010), decreasing rainfall and runoff (Sperna Weilang et al., 2021), and the expected growth of coastal populations (Neumann et al., 2015). In addition to global warming, climate projections predict an increase in the number of summer heat wave days: these climatic extremes are particularly worrying in terms of the tolerance of biological communities such as fish and the vulnerability of

macrotidal estuaries to deoxygenation.

The objective of this study is to provide a first assessment of the dissolved oxygen dynamics in the Charente estuary, one of the small estuaries of the Bay of Biscay:. The objectives were to determine the occurrence and extent of a possible estuarine oxygen minimum zone (eOMZ) and whether hypoxia could be triggered there. We used a complementary strategy based on a combination of longitudinal studies along the estuary axis and a few instrumented sites to determine the intensity and spatial

extent of potential deoxygenation. It focused on the summer period when low discharge, the presence of the turbidity maximum zone and summer heat are likely to promote low dissolved oxygen.

## 2 Material and method

### 2.1 The Charente estuary

Located on the French Atlantic coast, the Charente estuary is a small, shallow, macrotidal estuary with an average depth of approximately 9 m around Rochefort. It is a partially mixed to well-mixed macrotidal estuary, with stratified conditions occurring at very high river discharge (Toublanc et al., 2016). The tides are semidiurnal with an average amplitude of 4.2 m, reaching up to 7 m during spring tides (Figure 1). The tidal influence is stopped 50 km upstream of the mouth by a dam at Saint-Savinien, which is opened during spring tides to prevent flooding of downstream areas (Modéran et al., 2012). The high tidal range and upstream tidal asymmetry and subsequent tidal pumping promote a turbidity maximum zone (TMZ), as observed in two large nearby estuaries, the Loire and the Gironde (Jalon-Rojas et al., 2017; Toublanc et al., 2016). Suspended particulate matter (SPM) concentrations are typically several grams of dry sediment per litre (Auguet et al., 2005). The location of the TMZ varies longitudinally between the river mouth and 30-40 km upstream, depending on river discharge and tidal cycles.

The Charente River is 360 km long and its water catchment area covers almost 10,549 km$^2$. The Charente River flows through the Charente estuary (Figure 1) into the Bay of Marennes-Oléron, the first oyster farming area in France (Goulletquer and Heral, 1997) and a major nursery ground for juveniles of the Bay of Biscay sole population (Le Pape et al., 2003; Modéran et al., 2012). There is a strong contrast between the densely populated coastal fringe (80 to 100 inhabitants km$^{-2}$) and the rural interior (40 to 60 inhabitants km$^{-2}$). The largest town in the estuary is Rochefort, with an estimated urban population of over 63,000. The rest of the catchment area is largely agricultural (75% of the surface area): vineyards, maize, mixed farming and livestock. For the period 2006-2022, the minimum, average and maximum daily flows of the Charente are 6.5, 65.2 and 584 m$^3$ s$^{-1}$ respectively (Table A1). The target low flow, defined as "the reference flow which allows good water status to be achieved and above which all uses are satisfied on average in 8 out of 10 years ", is 15 m$^3$ s$^{-1}$. In fact, flows below this threshold are recorded almost every year, demonstrating that the Charente has experienced severe low flows in recent decades, making this estuary particularly sensitive to water quality problems. Managing water during these critical periods to meet socio-economic needs, but also to protect migratory fish is therefore a major challenge (EPTB Charente, 2023).

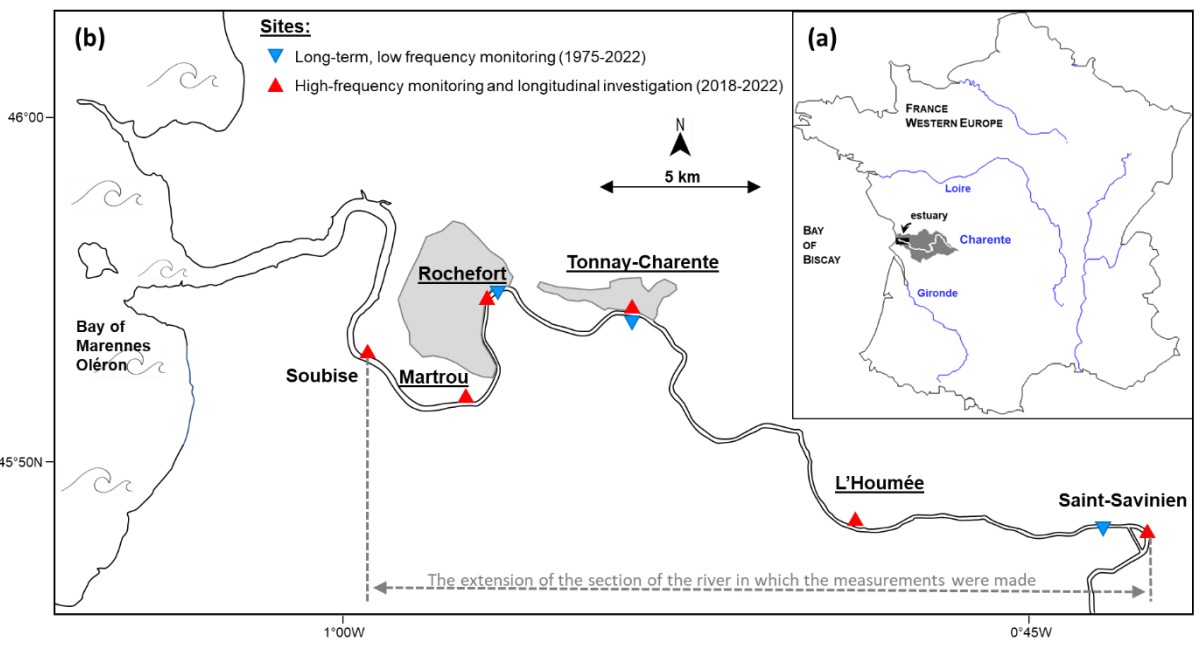

**Figure 1: (a) The Charente river between the Loire and Gironde rivers in western France: the water catchment area and the estuary of the Charente are highlighted by the grey and dark area.. (b) Location of the different instrumented sites along the Charente estuary. The arrows indicate the extent of the study area. The underlined city names correspond to the sites where hypoxia levels were measured during the study.**

**2.2 Data sources**

The study of dissolved oxygen variability in the Charente estuary is based on three different datasets (Table 1). The first two datasets were acquired between 2018 and 2020, based on a strategy previously developed in the Loire and Gironde estuaries
(Schmidt et al., 2019). The principle is to combine instrumented sites and longitudinal investigations. In the course of our work, we discovered the existence of an institutional low-frequency long-term monitoring of water quality, started in 1975, whose data were also used.

2.2.1 High-frequency summer monitoring of the Charente estuary

In order to better document the DO variability during the summer, the selection of a site to monitor was based on the experience
gained in the Loire and Gironde estuaries (Schmidt et al., 2019), with the hypothesis that a minimum oxygen zone could develop in the inner estuary, just upstream of the urban area of Rochefort. The first site was therefore located in Tonnay-Charente, 6.1 km upstream of Rochefort and 26 km (PK26) from the mouth of the estuary (in the following, the kilometre distance from the mouth is indicated as PK followed by the value). HOBO temperature/oxygen (optode) (-5 to +40°C; ± 0.2°; 0-20 mg L$^{-1}$; ±0.2 mg L$^{-1}$ < 8 mg L$^{-1}$ and ±0.5 mg L$^{-1}$ > 8 mg L$^{-1}$) and salinity/conductivity (0.05 - 36 µS cmto +40°C; ± 3%)
sensors with antifouling protection were used for a period of three weeks in summer 2018 to record temperature, salinity and dissolved oxygen every 15 minutes. The sensors were deployed on a floating pontoon at a depth of approximately 0.5 m below the surface. Sensor calibration was checked immediately before and after deployment. In summer 2019, Tonnay-Charente was instrumented again, complemented by a site further upstream at L'Houmée (PK37). In summer 2020, the downstream sector was instrumented with the same sensors at two sites, Rochefort (PK20) and Martrou (PK16). In addition, to test the possibility
of making it a permanent site, Tonnay-Charente was equipped with a SAMBAT multisonde (NKE Instrumentation, France) from the end of April to November 2020. The measured parameters are conductivity/salinity (salinity range 0.1 -42; ± < 5%), temperature (-5 to +35°C; ± < 1%), dissolved oxygen (0 - 20 mg L$^{-1}$; ±0.1 mg L$^{-1}$) and turbidity (Turner Designs CYCLOPS-7; 0 - 3000 NTU; ± < 5%). Turbidity (NTU) was converted to suspended particulate matter concentration (g L$^{-1}$) by establishing a relationship between turbidity and SPM in the Charente estuary. As part of a preliminary study, the advantage of placing the
sensors on floating pontoons is to limit the cost of the study, as they are accessible without nautical equipment. However, it should be noted that macrotidal rivers such as the Charente, can cause pontoons to become grounded during low tide, especially during spring tides when they are located near the shore. This problem was observed at the L'Houmée, Rochefort, and Matrou, resulting in gaps in the time series at these three sites.

**Table 1 Details of available data per site: site name, distance from estuary mouth (PK, in km), depth range (m), type of dataset, measurement period and link to the data repository. Sites are listed from upstream to downstream. The measured parameters are salinity, temperature, dissolved oxygen and suspended particulate matter (where * is indicated). There are three datasets: LF: low-frequency (6 – 12 measurements per year); HF: high-frequency (1 measurement every 10 to 20 minutes); LS: longitudinal survey. [a] accessed on 3 June 2023. [b] accessed on 22 July 2023**

| Site | PK Km | Depth m | Type | Measurement periods | Data repository |
|---|---|---|---|---|---|
| Saint-Savinien | 47.6 / | 1-5 | LF | 2011-2022* | https://adour-garonne.eaufrance.fr/station 05006860 [a] |
| L'Houmée | 37.0 | 0.3-6 | HF | 29 June to 7 Sept. 2019* | doi.org/10.17882/95886 [b] |
| Tonnay-Charente | 25.9 | 1.5-7.5 | LF HF | 2011-2022* 19 July to 9 August 2018; 29 June to 7 Sept. 2019; 24 April, to 16 Nov. 2020* | https://adour-garonne.eaufrance.fr/station 05002350 doi.org/10.17882/95886 [a] |
| Rochefort | 19.8 | 0.5-6.5 | LF HF | 1975-2022* 7 July to 7 Sept 2020 | https://adour-garonne.eaufrance.fr/station 05001500 [a] doi.org/10.17882/95886 [b] |
| Martrou | 16.1 | 0 - 6 | HF | 22 July to 19 August 2020 | doi.org/10.17882/95886 [b] |
| Saint-Savinien to Soubise | | | LS | August 9 2018 | doi.org/10.17882/95886 [b] |
| Tonnay-Charente to Soubise | | | LS | August 8 2019 | doi.org/10.17882/95886 [b] |

### 2.2.2. Longitudinal investigation of the Charente estuary

A complementary approach was to carry out longitudinal studies, corresponding to transects from one site to another following the axis of the estuary in order to study the spatial distribution of dissolved oxygen. In 2018, this was done on 7 August 2018 between Soubise (PK12) and Saint-Savinien (PK47) from the pontoons. At each station, the sensors were placed about 0.5 m below the surface for about 15 minutes to ensure that the sensors were in equilibrium with the water. In 2019, a first transect was carried out on 23 and 24 July, allowing us to measure dissolved oxygen at low and high tide at L'Hounée. Then, on 7 August, a second transect between Soubise and Tonnay-Charente was made from a boat with a SAMBAT multisonde permanently submerged at a depth of 0.5 m below the surface and recording data every 2 minutes. The 2018 longitudinal transect covered the longest distance along the estuary, about 35 km from Soubise, near the mouth, to Saint-Savinien. The 2019 transects have been reduced in upstream extension in the light of the 2018 observations.

### 2.2.3 Low-frequency long-term monitoring

The French Water Agency carries out long-term, low-frequency monitoring of surface water quality (0.5 m) in the Charente estuary. The low-frequency monitoring follows the guidelines of the European Water Framework Directive for the assessment of the ecological status of transitional waters. It includes the measurement of a range of chemical species such as organic and metallic pollutants, as well as temperature, SPM, salinity and dissolved oxygen. All data are freely available on a dedicated website (SIE Adour Garonne, 2023). There are three water quality stations in the Charente estuary. Rochefort (station number 5001500) has been monitored since 1975, initially with 8 visits per year, then from 1982 with 10 to 12. The extension of monitoring to Tonnay-Charente (station number 5002350) and Saint-Savinien (station number 5006860) is more recent, in 2011, with a lower frequency (6 visits per year).

### 2.2.4 Water quality criteria by DO values

The Water Framework Directive (WFD, 2000/60/EC) aims to achieve 'good status' for all ground and surface waters (rivers, lakes, transitional waters and coastal waters) in the European Union. The Directive identifies dissolved oxygen (DO) as one of the "five general chemical and physico-chemical elements supporting the biological elements". In this work we use the classification proposed by Best et al (2007) for water status according to DO levels: bad when DO levels are below 2 mg L$^{-1}$, poor to moderate when they range between 2 and 5 mg L$^{-1}$, and good when they are above 5 mg L$^{-1}$. Such management issues in estuaries justify the use of mg L$^{-1}$. The correspondence between mg L$^{-1}$ and µmol L$^{-1}$ is as follows: 1 mg L$^{-1}$ corresponds to 31.3 µmol L$^{-1}$, 2 mg L$^{-1}$ to 62.5 µmol L$^{-1}$ and 5 to 156.3 µmol L$^{-1}$.

### 2.2.5 Path analysis

Partial least squares structural equation modelling (PLS-SEM) was used to quantify the relationship between potential controlling variables and SPM and DO concentrations. Path analysis was performed using SmartPLS4 software (Ringle et al., 2022). The strength of the relationships between variables is represented by a path coefficient (β). The path coefficient is equivalent to the standardised regression coefficient. β indicates the direction and strength of the relationship between the explanatory variables and the explained variable. We used the following variables: discharge, tidal range, salinity, temperature, SPM and DO.

## 3. Results and discussion

### 3.1 Long-term, low-frequency monitoring of dissolved oxygen

The low-frequency dataset acquired at Rochefort covers 48 years, corresponding to very contrasting hydrological and climatic conditions. During this period, river discharge varies between 7.3 and 479 m$^3$ s$^{-1}$, covering the whole range of discharge in the Charente. As river discharge decreases, salinity increases from 0.25 to 17.8 (Figures 2a & 3a; Table A2). The highest salinities are usually observed in summer and autumn, during the low water period of the Charente. SPM concentrations show a wide

range of variability, from 0.002 to 12 g $L^{-1}$, i.e. more than 4 orders of magnitude (Fig. 2b & 3e). The lowest values are observed in winter, then there is an increase in SPM to several g $L^{-1}$ throughout the seasons. As for salinity, the highest SPM values are usually observed in summer and autumn. The turbidity maximum zone (TMZ), corresponding to an SPM concentration greater than 1 g $L^{-1}$, is present at Rochefort at least from July to November.

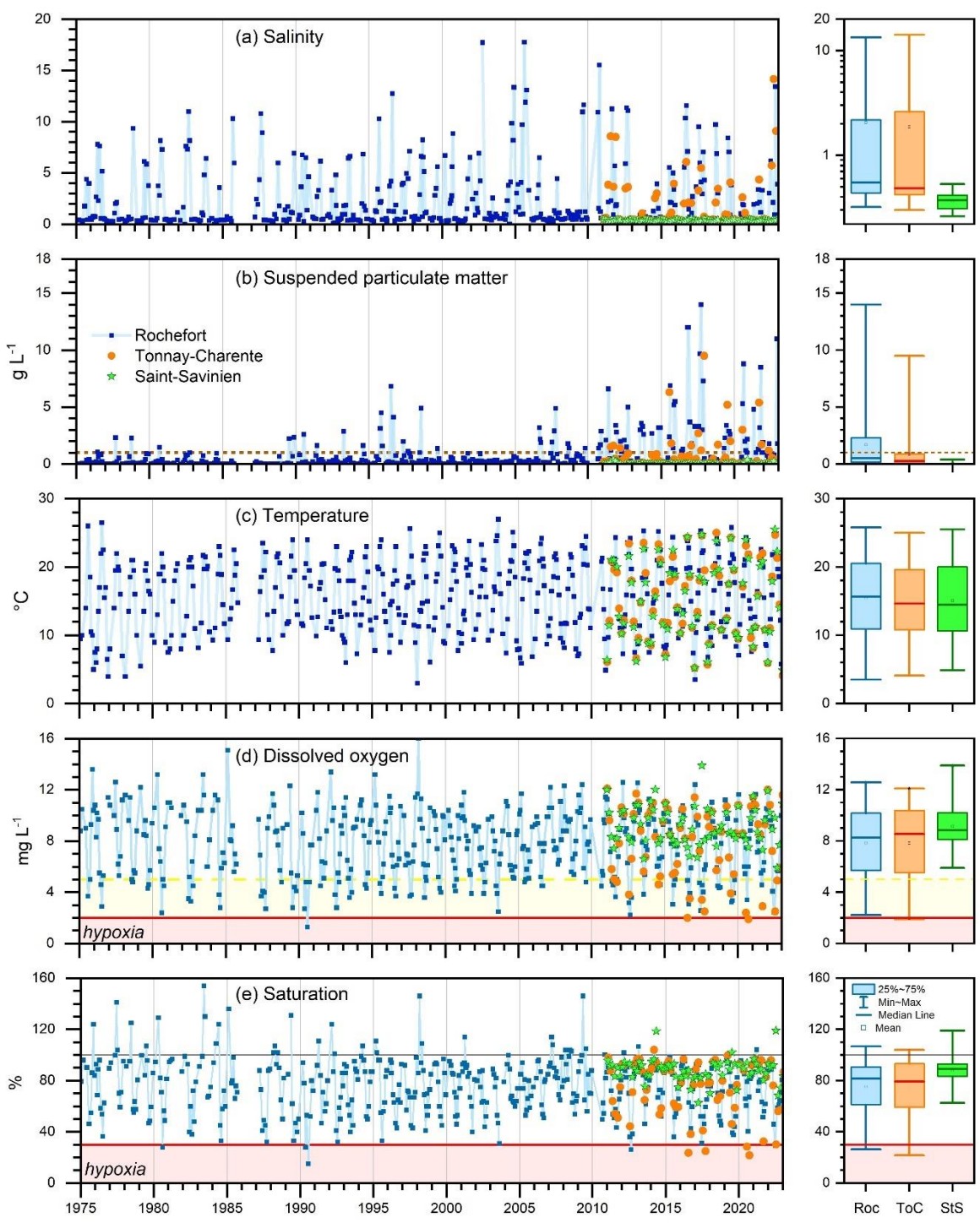

**Figure 2: Low frequency records of salinity(a), suspended particulate matter (b), temperature (c) and dissolved oxygen concentration (d) and saturation (e) in surface waters (0.5 m) at Rochefort for the period 1975-2022 and at Tonnay-Charente and Saint-Savinien for the period 2011-2022. The panel on the right shows the box plot (min, max, median, mean, 25%-75%) of each parameter per station (Roc: Rochefort; ToC: Tonnay-Charente; StS: Saint-Savinien) for the period 2011-2022. The dashed line in Fig. 2b shows**
**the SPM threshold of the TMZ (SPM > 1 g $L^{-1}$). The dashed and bold lines in Fig.2d show the DO threshold of 5 and 2 (hypoxia) mg $L^{-1}$, respectively. The thin and bold lines in Fig. 2e show the saturation threshold of 100 and 30 (hypoxia) %, respectively.**

Water temperature is also highest in summer, typically between 20.5 and 27.0°C, while in winter it ranges between 3 and 10.2°C (Fig. 2c & 3b). DO concentration and saturation show a strong seasonal signal (Fig. 2d/e & 3c/d). The highest values

are observed in winter and early spring, with concentrations and saturation usually above 10 mg L$^{-1}$ and 90 %. This period coincides with the lowest water temperature and negligible levels of salinity and SPM. There are even a few occasions in spring when there is strong oversaturation, up to 154 %, which may be associated with the spring bloom, as already reported in turbid estuaries (Goosen et al, 1999). Then, from June to October, the Rochefort estuarine waters of are always undersaturated as the water is the warmest, saltiest and most turbid. August is the month of consistently low dissolved oxygen, with minimum monthly saturations and concentrations of 53% and 4.6 mg L$^{-1}$ respectively. However, critical DO values are also observed in July, with the lowest value measured by the low frequency survey being 1.3 mg L$^{-1}$ (15% of saturation) on 23 July 1990. As the water cools in autumn, although still turbid, oxygenation gradually improves, with DO reaching values > 5 mg L$^{-1}$ from October onwards and returning to a situation close to saturation.

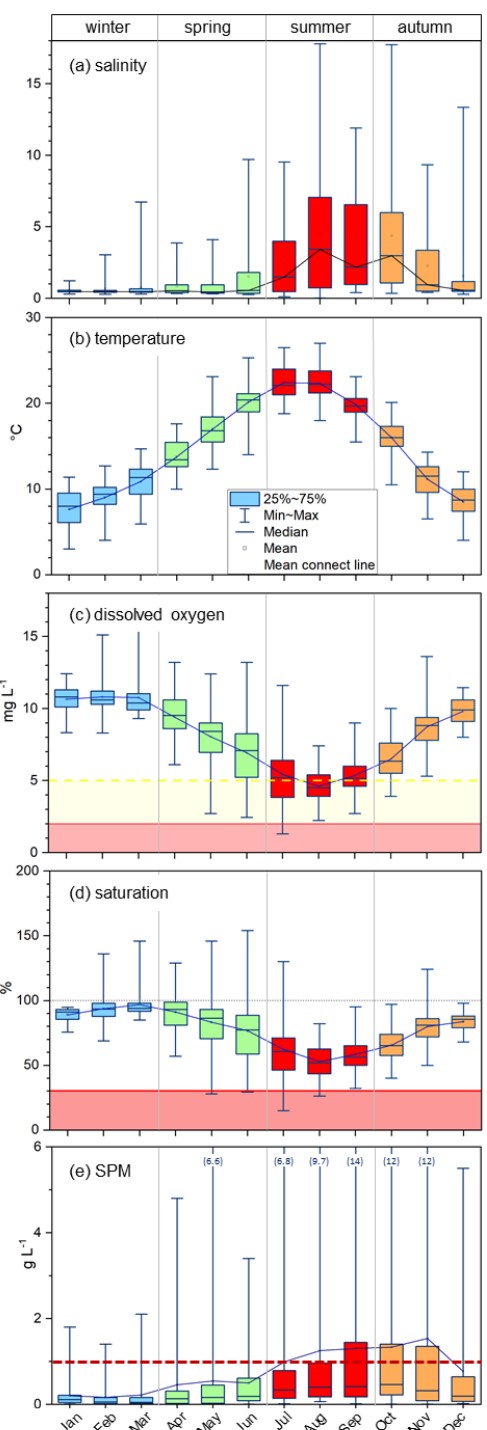

**Figure 3: Monthly box plots of (a) salinity, (b) temperature, dissolved oxygen (c) concentration and (d) saturation, and (e) suspended particulate matter (SPM) based on the long-term low-frequency monitoring at Tonnay-Charente for the period 1975-2022. In Figures 2c and 2d the DO thresholds of 5 mg L$^{-1}$ and 2 mg L$^{-1}$ or 30%, and 100% are indicated by dashed, bold and dotted lines**

respectively. **Grey vertical lines separate the seasons. The dashed line in Fig. 2e shows the SPM threshold of the TMZ (SPM > 1 g L$^{-1}$).The numbers in brackets (Figure 3e) correspond to the maximum SPM value for the months considered.**

About 6 km upstream, at Tonnay-Charente, the water quality station was added much later, starting in 2011 (Figure 2 ; Table A2). Not surprisingly, the salinity range is slightly lower than at Rochefort, from 0.3 to 14.2. Saline intrusion (salinity > 1) is usually observed in summer and autumn. Temperatures are quite similar, with a general monthly mean temperature difference of less than 0.5°C between Tonnay-Charente and Rochefort. Dissolved oxygen concentrations are also highest in winter and spring, ranging from 8.3 to 12.4 mg L$^{-1}$ and being close to the saturation equilibrium (95-104%). Spring is a transitional period, after which there is a sharp deterioration in water oxygenation. In July and September, the average monthly DO for the period 2011-2022 is 4.4 and 4.7 mg L$^{-1}$, with extremes ranging from 1.9 to 5.1 mg L$^{-1}$. In November, DO concentrations return to values above 5.9 mg L$^{-1}$, although undersaturation remains (57-93%) as the Charente discharge increases and the water is cooler. While the lowest minimum DO recorded at Tonnay-Charente (3.8 mg L$^{-1}$) was observed on 27 July 2012 during the period 2011-2015, there is a trend towards concentrations below 3.5 mg L$^{-1}$ almost every year since 2016 (Figure 2d). Unfortunately, the small number of measurements (6 per year) does not allow conclusions to be drawn on a possible trend.

The seasonal pattern of DO variability is similar between the stations of Rochefort and Tonnay-Charente, 6 km apart, both located in the turbidity maximum zone from spring to autumn. On the other hand, at Saint-Savinien, located about 28 and 22 km upstream of the other two stations, the low-frequency survey shows a large difference in salinity and dissolved oxygen (Figure 2; Table A2). This water quality station is located in the fluvial Charente estuary, as confirmed by the negligible salinity throughout the year (< 0.53). The main difference concerns DO concentrations, which range from 5.9 mg L$^{-1}$ to 13.9 mg L$^{-1}$ (saturation between 63 and 168 %), indicating good conditions for biological communities regarding the parameter.

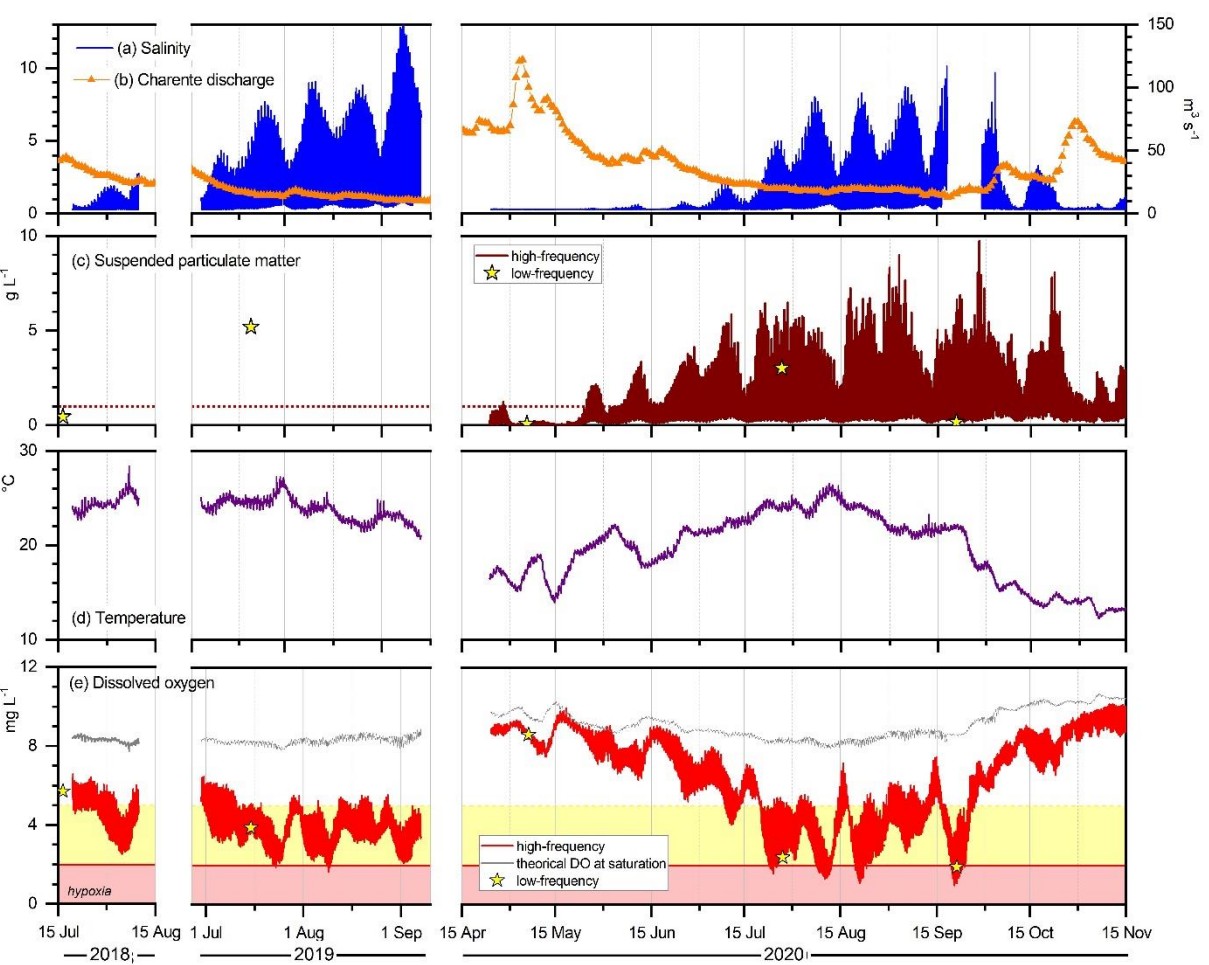

**Figure 4: High-frequency record of (a) salinity, (b) suspended particulate matter, (c) temperature and (d) dissolved oxygen in the surface waters of Tonnay-Charente during the summers of 2018 and 2019, and from spring to autumn 2020. The Charente river discharge (b) is also shown in the upper panel. Details of the instrumented periods are given in Table 1. The dashed line in Fig. 2b shows the SPM threshold of the TMZ (SPM > 1 g L$^{-1}$). The theoretical dissolved oxygen concentration at saturation, calculated from temperature and salinity, is plotted along the measured DO concentration. The dashed and bold lines indicate the threshold of 5 and 2 mg L$^{-1}$, respectively.**

Long-term, low-frequency monitoring highlights that severe summer deoxygenation occurs in the inner Charente estuary near and upstream of the town of Rochefort. Values below 3 mg L$^{-1}$ were measured several times from 1975 to 2022, with no a clear trend over decades. Given the low number of measurements (6 to 12 per year, depending on the water quality station concerned), it is not possible to estimate whether the lowest values have been reported and the duration of possible hypoxic events.

### 3.2 High-frequency monitoring of dissolved oxygen

### 3.2.1 DO dynamic at Tonnay-Charente

The first high-frequency record at Tonnay-Charente covered three weeks, from 17 July to 9 August 2018, when the Charente River entered a low-water period with a decrease in discharge from 45 to 23 m$^3$ s$^{-1}$ (Figure 4). At the same time, a short heat wave occurred at the beginning of August, causing a rapid increase in temperature, with a maximum of 28.4°C, and highlighting the high variability of dissolved oxygen. In this context, it can be seen that dissolved oxygen was already highly undersaturated at the end of July, with the difference between measured and theoretical DO being around 2.3 and 3.8 mg L$^{-1}$ depending on the tidal moment. Then, coinciding with the rise in temperature, which peaked on 6 August, there was a rapid decrease in dissolved oxygen, with daily concentrations falling from 5.4 to 3.1 mg L$^{-1}$ (63.3% to 37.9%) in a few days. The DO deficit reached 5.7 mg L$^{-1}$, indicating an increase in DO consumption. There was then a recovery of DO with the decrease in temperature to reach almost 5 mg L$^{-1}$ at the end of the summer 2018 observation. The minimum DO at the beginning of August 2018 is 2.5 mg L$^{-1}$, just above the hypoxic level. However, if we consider the threshold of 5 mg L$^{-1}$ for daily mean DO as good for the ecosystem (Best et al., 2007), 13 of the 22 observed days were below this criterion.

In 2019, the instrumentation period was 70 days, with an early implementation date of 29 June 2019 due to a first heat wave at the end of June. The low water period was particularly severe in the summer of 2019, with an average river discharge of 12 m$^3$ s$^{-1}$ (July to September) compared to 25 and 17 m$^3$ s$^{-1}$ in 2019 and 2020 (Figure 4). The very low discharge of the Charente in 2019 was reflected in much higher salinities in 2019 compared to 2018. Semi-diurnal tidal and spring/neap cycles were clearly visible in the salinity signal. Temperature variations were influenced by the different warm periods and heat waves. DO concentration was also highly variable, influenced by the tidal cycle, which was outweighed by the effect of warm periods, during which minima as low as 1.65 mg L$^{-1}$ were measured. From 11 July 2019 dissolved oxygen was below 5 mg L$^{-1}$ most of the time. On 23 July and 9 August 2019, the concentration even reached very critical levels, below the hypoxic threshold (2 mg L$^{-1}$) and an oxygen deficit of more than 6 mg L$^{-1}$. 69 of the 71 observed days were below 5 mg L$^{-1}$, but always above 2 mg L$^{-1}$.

In 2020, still at Tonnay-Charente, monitoring started at the end of April when the Charente discharge decreased to levels that allowed TMZ to develop and persist until the autumn floods (Fig. 4). SPM concentrations varied from 0.1 to 8 g L$^{-1}$ during the summer (Fig. 4c). SPM showed the presence of TMZ when the Charente discharge was less than 70 m$^3$ s$^{-1}$, from mid-May to November. The autumn cooling in 2020 was accompanied by a rapid recovery of oxygenation to higher levels (i.e. > 5 mg L$^{-1}$), although SPM concentrations are still high (> 1 g L$^{-1}$). From June to September 2020, representing a total of 92 days, there were 74 and 2 days, respectively, where the daily mean DO was below the threshold of 5 and 2 mg L$^{-1}$.

The extended monitoring period supports the observations from the low-frequency monitoring, with DO concentrations approaching equilibrium in spring and autumn and hypoxic events in summer. The periods during which hypoxia levels have been measured are usually short, typically between 3 and 6 days, which calls into question the frequency of measurements to capture them. The low frequency measurements have been plotted along the high-frequency records (Fig. 4). A first comment

is the good agreement between the two datasets. One might have expected a problem with the quality of the low-frequency monitoring data, which is carried out by operators rather than scientific teams. The agreement between the two datasets validates the use of these data, which are often discarded by research teams. However, the low-frequency data cannot capture
large variations in DO, which is expected with only two measurements taken in summer at the water quality station of Tonnay-Charente (in July and September). On the basis of the low-frequency monitoring, the water quality in 2018 and 2019 would be described as relatively good, i.e. DO > 5 mg L$^{-1}$, in terms of dissolved oxygen. In 2020, coincidentally, both measurements were taken when dissolved oxygen was at hypoxic levels. But these comparisons illustrate, if proof were needed, that high-frequency monitoring is essential in macrotidal estuaries. Furthermore, comparing the 2018-2020 high-frequency survey with
the long-term, low-frequency AEAG monitoring does not allow us to determine whether these severe deoxygenations have changed significantly since 1975.

### 3.2.2 DO dynamic apart Tonnay-Charente

A second site was instrumented in 2019 at L'Houmée (PK37), halfway between Tonnay-Charente (PK25.9) and Saint-Savinien (PK47.6). The data show the same trends as those for Tonnay-Charente, but with attenuation in the case of salinity due to the
more upstream location: salinity never exceeds 2. Dissolved oxygen also shows a temporal variability similar to that of Tonnay-Charente, but with a slightly wider range of values (Figure 5a). During the tidal cycle, minimum values, typically between 2.1 and 2.4 mg L$^{-1}$ (24.4-30.4%), are always observed at high tide, indicating the advection of oxygen-poor water from downstream (Figure 6). As a result, the minimum DO is always equal to or higher than the values recorded at Tonnay-Charente. Conversely, at low tide, the DO range at L'Houmée is much higher between 4.7 and 7.9 mg L$^{-1}$ (58-92%) (Figures 5 & 6). This suggests
advection of oxygenated water from upstream, confirmed by the low frequency monitoring at Saint-Savinien, about 10 km upstream of L'Houmée (Fig. 2d): DO at Saint-Savinien is always higher than 5.9 (Fig.1 and 2d). As a result, the daily mean DO is higher in L'Houmée than in Tonnay-Charente, with only half of the observation days having a daily mean of less than 5 mg L$^{-1}$ (most of them in Tonnay-Charente).

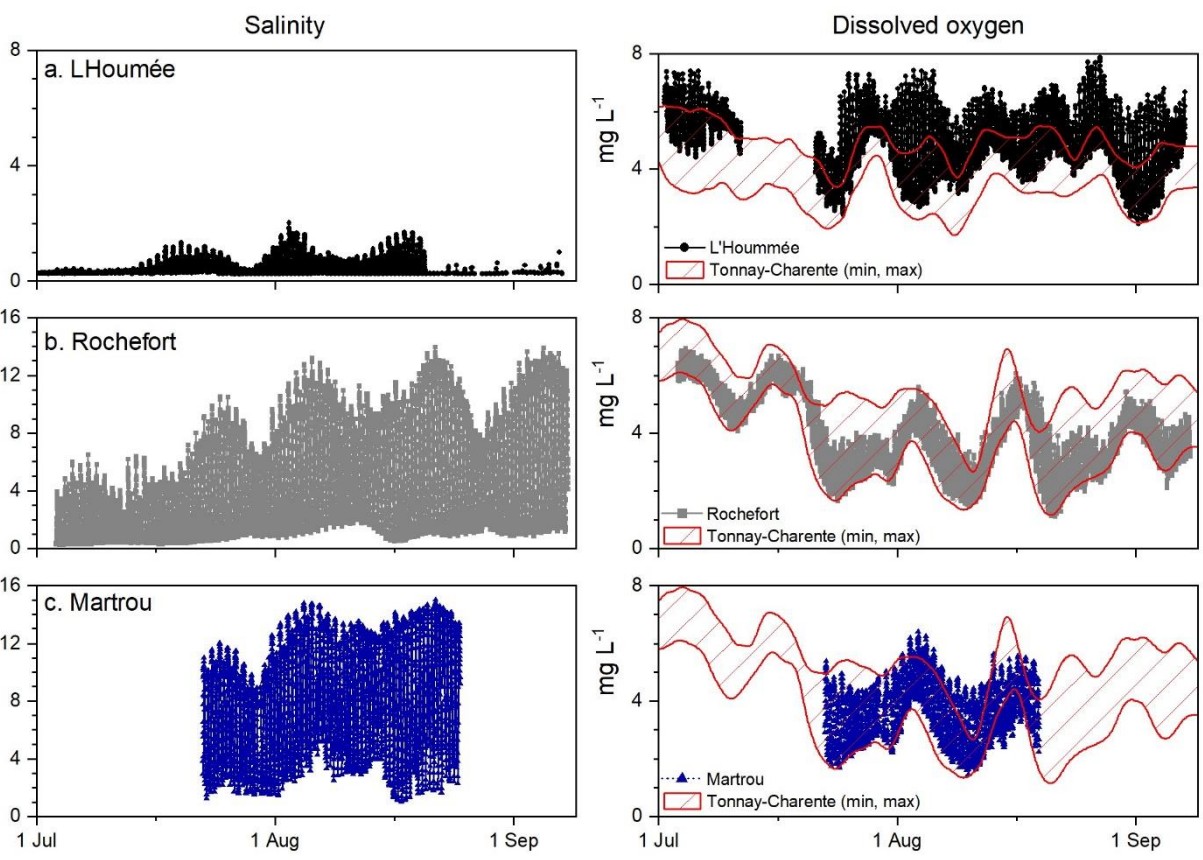

**Figure 5: High-frequency record of salinity and dissolved oxygen in the surface waters of L'Houmée during summer 2019 (a) and Martrou and Rochefort during summer 2020 (b). The summer DO range (minimum, maximum) at Tonnay-Charente is shown for comparison.**

In 2020, two stations were instrumented in the downstream sector between Soubise (KP12.2) and Tonnay-Charente (KP25.9)

at Matrou (KP16.1) and Rochefort (KP19.8). Matrou has a slightly higher salinity than Rochefort due to its position further downstream and closer to the mouth, but both are higher than Tonnay-Charente: maximum salinities are 15.0, 14.4 and 7.7 respectively (Figures 5 & 6). The DO concentration in Rochefort and Tonnay-Charente shows limited differences (Fig. 5). In summer, the minimum DO values are usually the same, but in September, when the maximum DO values are higher in Tonnay-Charente, Rochefort appears as the station with the lowest DO. Martrou was monitored for a shorter period, and its minimum

values are similar to those of the other stations. Only briefly, during high tide, DO could be 1 to 2 mg L$^{-1}$ higher than at the other two stations due to the advection of more oxygenated seawater. In short, the waters of Rochefort always showed the lower maximum values, which could be explained by the fact that the station is surrounded by oxygen-poor waters, whereas Tonnay-Charente and Martrou could benefit from the advection of fluvial or marine waters with a higher DO concentration, depending on the Charente discharge and the tidal range.

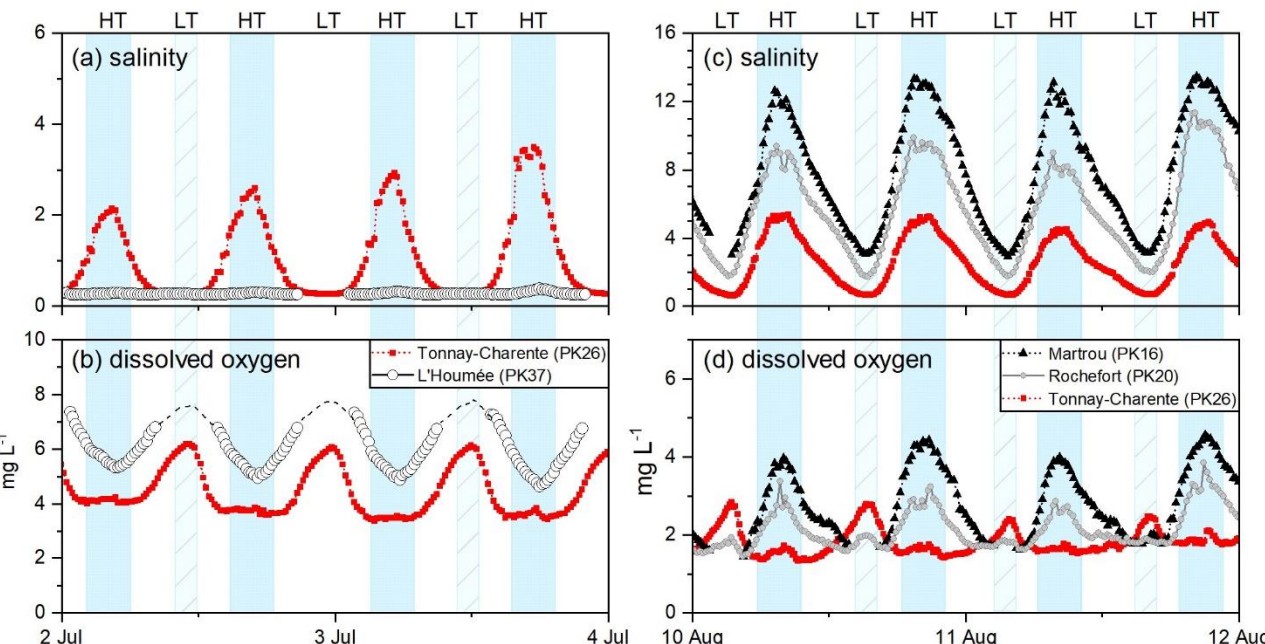

**Figure 6: Comparison of changes in salinity and dissolved oxygen with the tidal cycle: left panel (a; b) at L'Houmé and Tonnay-Charente on 2 and 3 July 2019 and right panel (c and d) at Martrou, Rochefort, Tonnay-Charente on 10 and 11 August 2020. PK corresponds to the kilometric distance of the site from the estuary mouth. LT and HT correspond to low and high tide, respectively;**

**the solid and dashed rectangles underline these periods. The dotted lines on the DO curve at L'Houmée (Fig. 6b) are an indication of the missing data due to the grounding of the pontoon at low tide.**

### 3.3 Location of the estuarine oxygen minimum zone

The identification of hypoxia in the Charente estuary then required an assessment of the location and horizontal extension of

the estuarine oxygen minimum zone (eOMZ), based on several approaches. Firstly, the low-frequency data set has already been used to define the maximum extent of the eOMZ. For the period 2011-2022, the mean DO at Saint-Savinien is 9.1 ± 1.5 mg L$^{-1}$ (87.9 ± 9.1%) and the lowest value recorded, in September 2022, is 5.9 mg L$^{-1}$ (62.7%) (Table A2; Fig. 2). Thus, at Saint-Savinien, in the fluvial estuary about 48 km from the mouth of the Charente, the oxygenation of the water remains well

above the hypoxic threshold (< 2 mg L$^{-1}$ or 30%). On the other hand, the two water quality stations in the inner estuary
(Rochefort and Tonnay-Charente) are clearly located in an area where severe deoxygenation has already been observed in
summer and early autumn. In Tonnay-Charente, high-frequency monitoring in 2018-2020 confirms the occurrence of low DO
levels in summer, as already shown by low-frequency monitoring since 2011. On the basis of these observations, it is already
possible to place an eOMZ in a sector between 19.8 and 25.9 km from the mouth, with an upstream extension of less than 47.6
km.

More recent instrumentation at sites upstream (L'Houmée) and downstream (Rochefort, Matrou) of Tonnay-Charente may
provide additional clues (Figure 5). Firstly, the high-frequency records confirm the presence of the eOMZ at Tonnay-Charente,
with strong deoxygenations and more frequent summer hypoxia than suggested by the low-frequency monitoring (Fig. 2, 3
and 4). Secondly, the low DO recorded at L'Houmée (PK37) at high tide shows that the upstream extent of the eOMZ is not
very far from Saint Savinien (PK47.6). In the lower reaches, the occurrence of severe deoxygenation at Rochefort and Martrou
(PK16.1) shows that the eOMZ extends downstream.

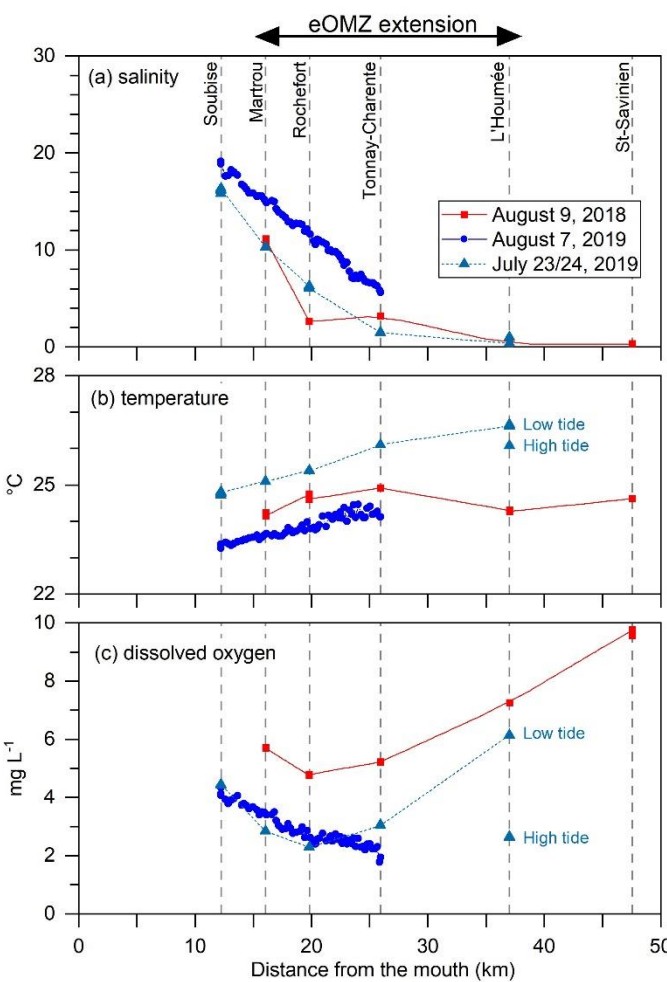

**Figure 7: Longitudinal distributions of (a) salinity, (b) temperature and (c) dissolved oxygen in surface water according distance from the mouth. The position of the instrumented sites is indicated by the dashed lines. The arrow shows a spatial extension of the eOMZ along the estuarine axis.**

To determine the spatial extent of deoxygenation, we applied a complementary strategy based on longitudinal surveys along
the estuary axis, as developed in the Gironde estuary (Schmidt et al., 2019) (Figure 7). The three longitudinal transects show
a concave profile, with minimum values in a sector between 16 and 37 km from the mouth, corresponding to a larger area than
that estimated on the basis of monitoring. On 23/24 July, L'Houmée (PK37) was visited twice, at low and high tide. At low
tide, DO was 6.14 mg L$^{-1}$, a high level explained by the arrival of oxygenated river water such as that observed at Saint-
Savinien, 10.5 km upstream. At high tide, however, DO drops to 2.65 mg L$^{-1}$, indicating the advection of DO-depleted estuarine

water. This site must therefore be close to the maximum upstream extent of the estuarine oxygen minimum zone. DO values are higher, typically > 4-5 mg L$^{-1}$, at low tide at L'Houmée and at high tide at Matrou, due to downward and upstream migration of well oxygenated fluvial and marine waters respectively. We can therefore assume that the boundaries of the eOMZ are close to these two sites. The eOMZ thus appears to be anchored around Rochefort, with an extension of several tens of kilometres upstream and less than 10 kilometres downstream of the town. It should be noted that the measurements were taken in surface water and that values close to anoxia are to be expected at the bottom of the water column. The spatial extent of the deoxygenated zone is critical for benthic species and fish, especially for migratory species (Hugman et al, 1984; Alabaster and Gough, 1986; Vaquer-Sunyer and Duarte, 2011; Woodland et al., 2009).

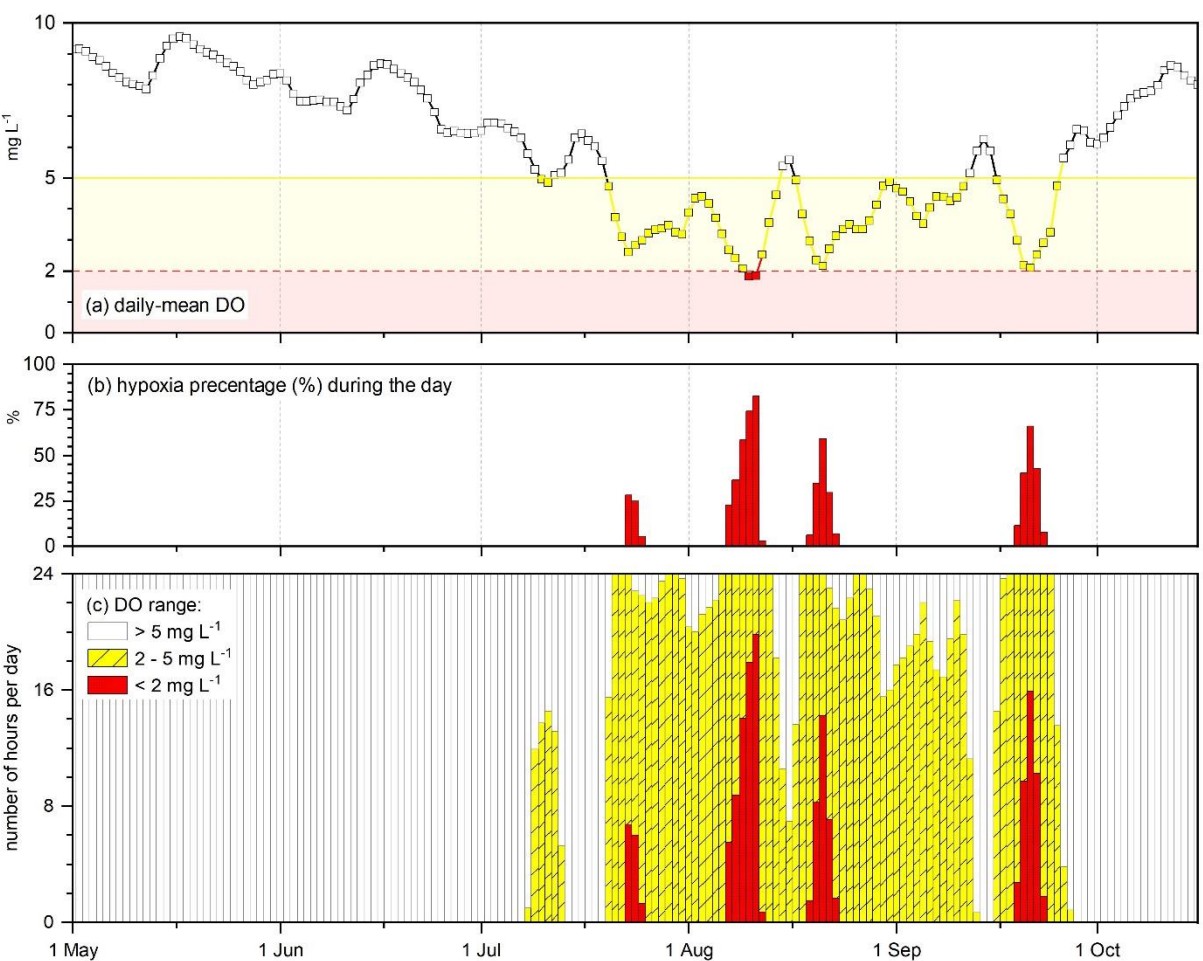

Figure 8: (a) daily-mean DO, (b) hypoxia percentage (%) during the day and (c) number of hours per day during which dissolved oxygen concentration of surface water in Tonnay-Charente is within the range of: < 2, 2-5, and >5 mg L$^{-1}$, from May, 1 to October, 15 2020.

## 3.4 Duration of hypoxic periods

All the data presented in this work show that the Charente estuary has a summer estuarine oxygen minimum zone with DO below 5 mg L$^{-1}$, with a spatial extent of about 25 km under current discharge and temperature conditions (Figure 7). The estuarine waters could reach borderline or even hypoxic levels during tidal cycles (Figures 4 & 5). Depending on species and stage of development, fish can tolerate hypoxic conditions for short periods (Curran and Henderson, 1988). Therefore, in addition to estimating the extent of the eOMZ, it is necessary to evaluate the duration of hypoxia to assess whether an estuary is of sufficient quality to maintain healthy conditions, particularly for migratory species (Hugman et al, 1984). For such an evaluation, we calculated the number of hours per day during which the DO concentration of the surface water in Tonnay-Charente was in the range of < 2, 3-5, >5-7 mg L$^{-1}$ in 2020, based on the high frequency monitoring, which has the interest of covering a large temporal period from spring to autumn. As described in section 3.2.1, DO levels are higher than 5 mg L$^{-1}$ until

the beginning of July and after the end of September. In summer, the daily average is generally between 2 and 5 mg L$^{-1}$, with only two days below 2 mg L$^{-1}$, on 10 and 11 August, when the water was hypoxic for 75-83% of the day. In the nearby Gironde

estuary, water managers have set a minimum daily average DO target of 5 mg L$^{-1}$ to significantly improve water quality for ecosystems and downstream migration of juvenile fish (Schmidt et al., 2017). According to these quality criteria, summer DO levels in Tonnay-Charente can already be considered insufficient. In addition, there are several periods where the daily mean DO is higher than 2 mg L$^{-1}$, but the measured DO is below the hypoxic threshold for 0.7 to 15.9 hours per day, i.e. 5.5 to 66.2% of the day (Figure 8). As a result, fish in the Charente estuary can be exposed to hypoxic conditions for several hours a day or

over several tens of kilometres in summer. The synthesis by Breitburg et al (2002) provides a comprehensive inventory of the effects of hypoxia on fish, which can lead to large reductions in the abundance, diversity and yield of fish in affected waters. Hypoxia and warming may also act synergistically, as hypoxia tolerance is generally lower in warmer waters, especially for larger fish (Sampaio et al., 2021; Verberk et al., 2022). Based on a review of the literature focusing on temperate estuaries, Arevalo et al. (2023) concluded that periods of low DO are generally episodic and do not necessarily pose a serious threat to

estuarine organisms if they occur for very short periods and do not recur. However, it is clear that the number of studies is limited. There is therefore a need for a better understanding of the thresholds required by biological organisms in estuarine waters.

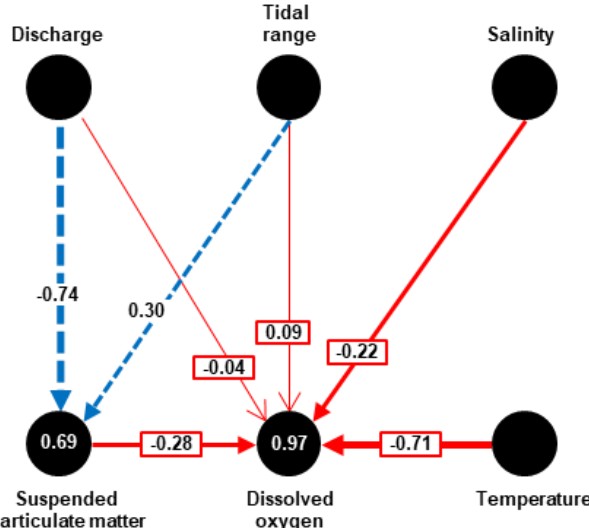

**Figure 9: Path analysis to define the relative importance of the factors controlling the temporal variability of dissolved oxygen (bold**
**lines) and of suspended particulate matter (dashed lines) based on the Tonnay-Charente 2020 dataset. The path analysis was performed with SmartPLS4 software (Ringle et al., 2023) using the following parameters: discharge, tidal range, salinity, temperature, SPM and dissolved oxygen. The numbers on arrows correspond to the path coefficient (β), its thickness to the strength of the relationship between the possible controlling factor and the explained variable. The numbers in dark circles (SPM, DO) correspond to R$^2$.**

**3.5 Hierarchy in forcing factors**

The restoration of good ecological status for transitional waters is an objective of the European Water Framework Directive (Best et al., 2007). Various measures could be implemented to mitigate hypoxia, and there are examples of improvements in European estuaries achieved by installing or renovating a wastewater treatment plant to reduce organic matter and nutrient inputs (Lajaunie-Salla et al., 2019). To optimise management strategies, modelling is an efficient tool to quantitatively test

potential solutions to mitigate hypoxia and provide guidance for setting targets to maintain good water quality in estuaries (Kemp et al., 2009; Lajaunie-Salla et al., 2019). However, such an exercise requires a good knowledge of the DO dynamics in the system under consideration, as estuarine deoxygenation is the result of a complex interaction of environmental factors.

The time series recorded in the Charente estuary show the high variability of DO and the occurrence of short and recurrent hypoxia in summer. For a first estimation of the controlling factor of DO variability using a path-analysis, we used only the

high frequency data set from 2020. This dataset also includes SPM concentrations in addition to temperature, salinity and DO

data (Schmidt and Diallo, 2023). It also covers a time period from April to November, which encompasses the DO deficit period and then the temporal variability of water quality parameters in different hydrological and meteorological contexts. Fig. Based on the path analysis, three groups of parameters can be distinguished according to their influence on DO. River discharge and tidal range have little direct influence ($\beta < 0.10$), but act indirectly by influencing the variability of salinity and SPM (Fig. 9). SPM (b=-0.28) and salinity (b=-0.22) have a negative influence. However, temperature appears to be the main contributor to DO variability ($\beta$=-0.71). The reason is that oxygen solubility decreases (Conley et al., 2009) and DO-consuming biogeochemical processes accelerate with increasing water temperature (Goosen et al., 1999). For the period 2018-2020, DO is always higher than 5 mg L$^{-1}$ when the water temperature was below 18°C, even when TMZ is still present. To date, the water temperature at Rochefort and Tonnay-Charente is always below this threshold from October to April. Given the expected temperature rise in south-west France, which could be as high as +4°C by 2100, the question arises as to the evolution of the timing of the onset of unfavourable temperatures. The Charente estuary is a corridor for migratory species such as allis shad, salmon and eel. The downstream migration of juvenile fish is most frequent in autumn, when present-day conditions are favourable for entering the estuary (EPTB, 2013). Just to the south, in the Gironde-Garonne-Dordogne estuary continuum, it has been shown that there is a limited period of favourable temperature and dissolved oxygen conditions for the European sturgeon breeding season (Delage et al., 2019). Continued increases in temperature and associated deoxygenation are expected to further reduce their survival, leading to a possible shift in their repartition northwards in the long term (Lassalle et al, 2010; Arevalo et al, 2023).

## 4. Concluding remarks

In short, the combination of historical low-frequency data and newly acquired high-frequency data has made it possible to highlight the occurrence of severe summer oxygen deficit, even hypoxia, in the Charente estuary. The estuarine oxygen minimum zone is located in the inner estuary, near Rochefort and Tonnay-Charente, and extends for 20-30 km. This work has demonstrated the need to monitor such a small estuary, which is effective from 2021, by extending a regional monitoring network initially developed in the Gironde estuary since 2004 (Etcheber et al., 2011). The evidence of summer hypoxia in the Charente estuary also underlines the need to study all the macrotidal and highly turbid small estuaries between the large Loire and Gironde estuaries.

In any case, it highlights the need to focus on small estuaries in order to understand and predict future changes in the face of anthropogenic and climatic pressures. Reductions in wastewater pollution in recent decades have contributed to improvements in estuarine DO. However, the trajectory of estuarine DO is difficult to predict and will depend on their vulnerability to the effects of climate change. A better understanding of the thresholds required by biological organisms in estuarine waters is also needed to develop appropriate monitoring, restoration and mitigation strategies. Temperature is a key parameter in macrotidal estuaries, which often have a turbidity maximum zone: such a feature promotes the occurrence of severe deoxygenation to hypoxia. At the scale of an estuary and its catchment, there are almost no levers to limit water temperature increases. Supporting low summer flows with water stored in reservoirs can reduce the residence time of estuarine water, increase the inflow of more oxygenated river water, and limit the TMZ concentration and hence the oxygen consumption associated with bacterial degradation.

## 5. Appendix A

460 **Table A1 Annual range (minimum, mean, maximum) of the Charente discharge for the period 2006-2022. Data are from https://hydro.eaufrance.fr/sitehydro/R4230010/fiche (last access the 2 June 2023) corresponding to station Chaniers [Beillant].**

| Year | minimum | mean | maximum |
|------|---------|-------|---------|
| 2006 | 9.8 | 65.3 | 380 |
| 2007 | 14.4 | 75.2 | 584 |
| 2008 | 15.5 | 68.1 | 227 |
| 2009 | 8.4 | 48.6 | 425 |
| 2010 | 9.1 | 53.7 | 184 |
| 2011 | 6.7 | 33.1 | 294 |
| 2012 | 6.5 | 63.4 | 414 |
| 2013 | 17.5 | 99.4 | 464 |
| 2014 | 18.3 | 99.1 | 453 |
| 2015 | 14.7 | 52.5 | 200 |
| 2016 | 12.5 | 80.2 | 507 |
| 2017 | 8.5 | 31.0 | 181 |
| 2018 | 13.8 | 101.4 | 508 |
| 2019 | 7.4 | 69.1 | 321 |
| 2020 | 11.2 | 65.5 | 263 |
| 2021 | 14.9 | 71.4 | 485 |
| 2022 | 7.3 | 32.0 | 148 |

**Table A2** Range of salinity, temperature (°C), dissolved oxygen, in concentration (mg L$^{-1}$) and saturation (%) for each water quality station whose data series is plotted in Figure 2. For Rochefort, calculations have been made for the whole dataset (1975-2022) and for the period 2011-2022 to allow comparison with the other two other stations which only cover the period 2011-2022. Bold and underlined numbers correspond to concentrations and saturation less than 3 mg L$^{-1}$ or 35% and greater than 10 mg L$^{-1}$ or 100% respectively.

| Site | Years | Month | N | Salinity | | | Temperature (°C) | | | DO (mg L$^{-1}$) | | | Saturation (%) | | |
|---|---|---|---|---|---|---|---|---|---|---|---|---|---|---|---|
| | | | | mean | min. | max. | mean | min. | max. | mean | min. | max. | mean | min. | max. |
| Rochefort | 1975-2022 | Jan | 23 | 0.6 | 0.3 | 1.2 | 7.6 | 3.0 | 11.4 | 10.7 | 8.3 | 12.4 | 89 | 76 | 95 |
| Rochefort | 2011-2022 | Jan | 13 | 0.5 | 0.4 | 1.1 | 7.3 | 3.5 | 11.3 | 10.9 | 8.3 | 12.4 | 89 | 76 | 95 |
| Tonnay-Ch. | 2011-2022 | Jan | 11 | 0.6 | 0.4 | 2.1 | 7.9 | 5.2 | 10.3 | 10.9 | 9.0 | 12.1 | 90 | 78 | 98 |
| St-SAvinien | 2011-2022 | Jan | 12 | 0.5 | 0.4 | 0.5 | 7.6 | 4.9 | 10.3 | 10.7 | 8.3 | 12.0 | 88 | 70 | 97 |
| Rochefort | 1975-2022 | Feb | 31 | 0.7 | 0.3 | 3.0 | 9.0 | 4.0 | 12.7 | 10.8 | 8.3 | 15.1 | 94 | 69 | 136 |
| Rochefort | 2011-2022 | Feb | 13 | 0.6 | 0.4 | 2.7 | 10.0 | 7.3 | 12.7 | 10.8 | 9.6 | 12.6 | 94 | 85 | 107 |
| Rochefort | 1975-2022 | Mar | 38 | 0.8 | 0.3 | 6.7 | 10.9 | 5.9 | 14.7 | 10.8 | 9.3 | 16.0 | 97 | 85 | 146 |
| Rochefort | 2011-2022 | Mar | 12 | 0.5 | 0.4 | 0.7 | 11.6 | 8.5 | 14.7 | 10.4 | 9.6 | 11.7 | 94 | 88 | 104 |
| Tonnay-Ch. | 2011-2022 | Mar | 11 | 0.4 | 0.4 | 0.6 | 11.9 | 9.6 | 13.5 | 10.5 | 9.6 | 11.0 | 96 | 88 | 104 |
| St-SAvinien | 2011-2022 | Mar | 11 | 0.4 | 0.4 | 0.4 | 11.6 | 9.1 | 13.2 | 10.0 | 6.7 | 10.8 | 90 | 63 | 97 |
| Rochefort | 1975-2022 | Apr | 44 | 0.8 | 0.3 | 3.9 | 13.7 | 10.0 | 17.6 | 9.4 | 6.1 | 13.2 | 91 | 57 | 129 |
| Rochefort | 2011-2022 | Apr | 12 | 0.6 | 0.3 | 1.9 | 14.7 | 11.4 | 16.7 | 9.1 | 7.1 | 10.5 | 90 | 72 | 99 |
| Rochefort | 1975-2022 | May | 41 | 1.0 | 0.3 | 4.1 | 17.0 | 12.3 | 23.1 | 8.0 | **2.7** | 12.4 | 83 | **28** | 146 |
| Rochefort | 2011-2022 | May | 12 | 1.0 | 0.4 | 3.7 | 17.4 | 15.1 | 21.8 | 7.6 | 4.1 | 9.0 | 78 | 46 | 92 |
| Tonnay-Ch. | 2011-2022 | May | 11 | 0.7 | 0.3 | 3.8 | 17.7 | 14.8 | 20.8 | 7.9 | 5.8 | 9.3 | 84 | 64 | 96 |
| St-SAvinien | 2011-2022 | May | 11 | 0.3 | 0.3 | 0.4 | 17.6 | 14.9 | 21.0 | 8.8 | 7.7 | 12.0 | 92 | 82 | 119 |
| Rochefort | 1975-2022 | Jun | 48 | 1.5 | 0.3 | 9.7 | 20.1 | 14.0 | 25.3 | 6.9 | **2.4** | 13.2 | 77 | **29** | 154 |
| Rochefort | 2011-2022 | Jun | 12 | 1.1 | 0.3 | 6.2 | 21.3 | 18.4 | 25.3 | 6.1 | **2.4** | 8.3 | 68 | **29** | 92 |
| Rochefort | 1975-2022 | Jul | 44 | 2.6 | 0.3 | 9.5 | 22.4 | 18.8 | 26.5 | 5.4 | **1.3** | 11.6 | 62 | **15** | 130 |
| Rochefort | 2011-2022 | Jul | 11 | 3.9 | 0.3 | 9.5 | 23.7 | 21.2 | 25.8 | 4.8 | 3.1 | 7.5 | 56 | 36 | 79 |
| Tonnay-Ch. | 2011-2022 | Jul | 11 | 2.6 | 0.3 | 8.6 | 23.2 | 19.6 | 25.0 | 4.4 | **2.0** | 8.5 | 50 | **24** | 96 |
| St-SAvinien | 2011-2022 | Jul | 12 | 0.3 | 0.3 | 0.3 | 23.1 | 20.1 | 25.5 | 8.5 | 6.6 | 13.9 | 99 | 76 | 168 |
| Rochefort | 1975-2022 | Aug | 42 | 4.7 | 0.3 | 17.8 | 22.3 | 18.0 | 27.0 | 4.6 | **2.2** | 7.4 | 53 | **26** | 82 |
| Rochefort | 2011-2022 | Aug | 12 | 4.8 | 0.5 | 11.4 | 22.8 | 20.2 | 24.9 | 4.2 | **2.2** | 6.6 | 48 | **26** | 73 |
| Rochefort | 1975-2022 | Sep | 47 | 4.2 | 0.4 | 11.9 | 19.8 | 15.5 | 23.1 | 5.3 | **2.7** | 9.0 | 58 | **32** | 95 |
| Rochefort | 2011-2022 | Sep | 12 | 4.1 | 0.4 | 11.6 | 20.2 | 17.9 | 22.5 | 5.1 | 3.6 | 7.6 | 55 | **39** | 82 |
| Tonnay-Ch. | 2011-2022 | Sep | 12 | 4.1 | 1.0 | 14.2 | 19.8 | 18.0 | 22.0 | 4.7 | **1.9** | 6.7 | 51 | **22** | 73 |
| St-SAvinien | 2011-2022 | Sep | 12 | 0.3 | 0.3 | 0.4 | 19.6 | 17.7 | 22.3 | 7.7 | 5.9 | 8.8 | 84 | 68 | 93 |
| Rochefort | 1975-2022 | Oct | 45 | 4.4 | 0.3 | 17.7 | 16.0 | 10.5 | 20.1 | 6.5 | 3.9 | 10.0 | 66 | 40 | 97 |
| Rochefort | 2011-2022 | Oct | 12 | 4.1 | 0.4 | 13.4 | 16.2 | 13.2 | 18.6 | 6.6 | 4.4 | 8.2 | 66 | 46 | 83 |
| Rochefort | 1975-2022 | Nov | 40 | 2.3 | 0.4 | 9.3 | 11.1 | 6.5 | 14.3 | 8.7 | 5.3 | 13.6 | 80 | 50 | 124 |
| Rochefort | 2011-2022 | Nov | 13 | 1.8 | 0.5 | 7.0 | 11.5 | 8.4 | 14.3 | 8.6 | 6.9 | 9.8 | 78 | 66 | 87 |
| Tonnay-Ch. | 2011-2022 | Nov | 11 | 2.7 | 0.4 | 9.1 | 11.7 | 9.1 | 14.5 | 8.6 | 5.9 | 10.7 | 79 | 57 | 93 |
| St-SAvinien | 2011-2022 | Nov | 11 | 0.4 | 0.4 | 0.5 | 11.4 | 8.8 | 14.0 | 9.0 | 7.7 | 9.8 | 82 | 73 | 89 |
| Rochefort | 1975-2022 | Dec | 35 | 1.6 | 0.3 | 13.4 | 8.5 | 4.0 | 12.0 | 9.8 | 8.0 | 11.4 | 84 | 68 | 98 |
| Rochefort | 2011-2022 | Dec | 11 | 1.5 | 0.5 | 4.3 | 8.2 | 5.8 | 11.1 | 10.3 | 8.9 | 11.3 | 86 | 78 | 93 |

## 6. Data Availability

Data are available on data repositories as detailed in Table 1

## 7. Author contribution

SS: Conceptualization, Methodology, Investigation, Data Curation, Date analysis; I.I.D. Investigation, Resources, Project administration. SS prepared the manuscript with contributions from the co-author.

## 8 Competing interest

The authors declare that they have no conflict of interest

## 9. Acknowledgment

This work was a contribution to the project QUEtSCHE (trajectoires de la QUalité des eaux et du fonctionnement des écosystèmes des EStuaires aquitains face au CHangEment climatique) funded by the OASU (Aquitaine Observatory of the Sciences of the Universe) and the AEAG (Agence de l'Eau Adour Garonne).

## 10. Special issue statement

The article is submitted to the special issue 'Low-oxygen environments and deoxygenation in open and coastal marine waters' edited by Grégoire M. et al

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
