# Peer review of "Title page"

_Biogeosciences, 2023_

## Author Comment (AC2)

In black : comments of RC2 / in blue reply of authors

I read the comments from Reviewer 1 and responses from the authors. I agree with the authors that this article provides a new and valuable dataset and raises attention to the deoxygenation situation of macrotidal estuaries. I also agree with Reviewer 1 that authors can further improve the clarity of statements and word choices (detailed in my specific comments) and include necessary information to help readers interpret results (e.g., SPM data to show the maximum turbidity zone, water depths of monitoring stations to justify using surface data alone). Reading the comments made by the two assessors made us realise that it was necessary to add information about the sites, macrotidal estuaries and the zone of maximum turbidity. This will be corrected in the revised version. We would like to thank the reviewer 2 for their careful reading and the suggestions and questions that followed. They will greatly improve the manuscript.

The turbidity maximum zone (TMZ) appeared on many occasions throughout the Results and Discussion section. However, there is no data or analysis on TMZ presented in the manuscript. According to Sections 3.1 and 3.5, the concentration of suspended particulate matter (SPM) is monitored at several stations and used to conduct path-analysis. However, the data of SPM is not shown anywhere. I suggest presenting the SPM data to directly support the analysis of TMZ and the link between SPM and oxygen dynamics. The reviewer is right, there are indeed measurements of the suspended matter load that were used for the analysis. The low-frequency monitoring carried out by the Water Agency includes this parameter. As the article focuses on dissolved oxygen, we did not include this parameter. Figure 2 will be modified to include the suspended matter load and the presence or absence of the maximum turbidity zone will be discussed for site depending of seasons.

The manuscript mainly presents and discusses variations of temperature, salinity, and dissolved oxygen (DO) in Charente. It suggests that temperature is the main controlling factor of DO. What are the conditions of other water quality parameters, such as inorganic nutrients, organic matter, and chlorophyll, in this estuary? Do they contribute to the low oxygen events in the Charente?
The analysis also suggests that the downstream estuarine water has a lower oxygen level while the upstream river water has a higher oxygen level. This spatial variability doesn't seem to be controlled by temperature. What are the other driving factors? In fact, few studies have been carried out in the Charente estuary, so little reliable data is available. This initial work was initiated following the constant occurrence of hypoxia in neighbouring macro-tidal estuaries. At this stage, the work has revealed summer hypoxia. The link with temperature is strong, because in autumn, although SPM is high, as soon as the temperature drops, oxygenation is restored to levels above 5 mg/L. For further interpretation, more detailed studies of particles (% carbon, lability) will be required. As far as chlorophyll is concerned, SPMs of several grams per litre exclude primary production.

In addition, Figure 4 shows that the measured oxygen level is way below the saturated oxygen level during summer. This suggests remarkable oxygen depletion, which can be related to the biological and chemical processes consuming oxygen. Are they due to microbial degradation of organic matter? While the oxygen-consuming processes are also affected by temperature, there should be sources of organic matter to fuel the microbial degradation. Therefore, I think other parameters in addition to temperature and salinity are also required to better understand the variations of DO. Indeed, in TMZ oxygen consumption is mainly related to microbial degradation, a process that is greatly influenced by temperature. Even if the carbon load is low, the huge charge of particles (several grammes per litre) implies an important oxygen consumption. The discharge was considered in the path-analysis as it influences directly water renewal, but its impact was low regarding the other considered parameters. The impact of discharge on dissolved oxygen will be added in Figure 8.

Specific comments
L31: the first oyster-farming area? → it will be corrected according the suggestion

L35: "salinity sensors"? Do you mean conductivity sensors? Yes, the HOBO Salt Water Conductivity/Salinity Data Loggers measure conductivity, and temperature, salinity is calculated https://www.onsetcomp.com/products/data-loggers/u24-002-c And what are used to measure temperature? HOBO Dissolved Oxygen Data Loggers measured dissolved oxygen and temperature.

L36: "will be" -> "is" → it will be corrected

L36: specify the sampling frequency of the "low-frequency" dataset → it will be precised

L37: "a degradation of oxygenation" -> "a deoxygenation trend" It is not exactly the idea, the sentence will be modify to clarify it.

L41: -> "main controlling factor of DO" → it will be corrected

L46: "diagnosing of" -> "diagnosing" → it will be corrected

L60: "less the attention" -> "less attention" → it will be corrected

L60: Not sure how we can conclude that "hypoxia in estuaries has received less the attention than in coastal waters". → just a rapid check: 775 results from Web of Science Core Collection for estuary/hypoxia (considering that some of the articles concern more the outer estuary) 922 + 174 for coastal water and continental shelf/hypoxia, or only 14 references in Gilbert and al 2010. Anyway, the sentence will be modified according this comment.

L67: "given their large impact on aquatic organisms" – can be deleted. It has been elaborated by the preceding sentences. → suggestion will be considered

L81-84: please provide references for these predicted changes. → relevant references will be added

L131: the depth of the selected sites should be described to help readers better interpret the data collected at 50 cm below the surface. Maybe mark them in Figure 1. → range of water depth will be added

L135: What parameters are measured through the multi-parameter sensor? → the same (temperature, conductivity and dissolved oxygen, and turbidity) it will be precised

L154: consistently use "50 cm" or "0.5 m" → it will done

L160: "Charente estuary is not known to have hypoxic situation" - is this statement based on monitoring or the geographic, physical, and/or chemical characteristics of the estuary that make it not prone to develop hypoxia? It is based on the absence of publication

L161-170: What methods are used to collect these low-frequency data? And at what depths are the measurements conducted? Are they comparable to those of the high-frequency data? These data are produced by certified laboratories that produced qualified data. Measurements are also made in surface waters. Details will be added

L175: delete "that" → it will done

181: "varies" -> "vary" → it will be corrected

187-188: This should be divided into two sentences. → it will done

L197: insert "with" before "DO reaching …" → it will done

L200: the unit of saturation in Fig. 2(d) is incorrect. The time period of the observations at Saint-Savinien is missing. …" → it will be corrected. Indeed, there is a mistake, Matrou will be replaced by Saint-Savinien

L217: delete "even" …" → the sentence will be modified to better convey the idea

L230: Fig. 4a. The shaded area of discharge is covered by that of salinity from time to time. Suggest plotting the discharge using a solid line on the top of salinity. …" → the figure 4a will be modified
Fig. 4d. The legend of the gray line could be modified as "DO at saturation". "DO-100%" can be misinterpreted as the "DO saturation – 100%" with a unit in percentage. …" → it is written in the figure caption, but it could be modified

L240: In Figure 4, where can we tell "the presence of a moderate TMZ"? → the presence of the TMZ is related to the fluvial discharge. Turbidity will be added in Figure 4.

L248: considering -> consider …→ it will be changed

L251, 257: will be -> was → it will be changed

L250-260: Please pay attention to the tense being used. It should be consistent throughout the manuscript. …→ it will be checked
L260: Please include the unit. → unit will be added

L262: "..allowed TMZ to develop and persist until the autumn floods (Fig. 4)." – this cannot be inferred from Figure 4. → as previously

L270: "This validates the use of these data …" -> "The agreement between the two datasets validates the use of the low-frequency data …"→ it will be corrected

L273: "inability of the low-frequency data to …"→ the low-frequency data will be added

L282: What trends of what variables? Figure 5 only shows dissolved oxygen. L295: Figure 5 – suggest marking the period of high and low tides on the time series. …"→ salinity will be added

L302: The previous paragraph (L284-286) suggests that at high tide, the advection of oxygen-poor water from downstream leads to low-oxygen condition. But this sentence suggests the opposite. L305: This contradicts L284-290 which suggests that the downstream estuarine water has a lower DO concentration while the upstream river water has a higher DO concentration → It depends of the site, this will be clarified

L310: "Oxygen" -> "oxygen"; Do you mean "localization" or "location" of the oxygen minimum zone? Location

L318-319: Do you mean the DO data collected in the summer of the 3 years shows a decreasing trend? The time series presented in Figure 4 is not obvious to support this statement. The data in 2018 only covers 3 weeks, which is not sufficient to represent the entire summer condition of 2018. → no, we want to say that DO decrease in summer as observed by low-frequency monitoring. The sentences will be clarified

L336-337: We need the information on the depths of these monitoring sites. Are there any data to show how stratified the water column is? …"→ there is no or limited stratification in macrotidal estuaries, especially in the upper section. It will be precised

L346: "reach borderline or even hypoxic, levels" -> "reach borderline or even hypoxic level" L353: delete "comprised" L355: delete the second → "," and "comprised" "target"will be suppressed

L354: add a "," after "conditions" L356: -> "summer levels of DO": → it will be added

L365: "However, it is clear that the number of studies is limited and rather dated. " – not clear to me how we can come to this conclusion. The cited references in the preceding sentences are mostly between 2021 and 2023. → Yes indeed and the most recent is Arevalo et al. (2023) that a review of the literature focusing on temperate estuaries. Therefore, the recent references do not rule out the fact that a large part of the works is not of recent date.

L382: "used the 2020 dataset, which has the interest to include also …" -> "used the dataset obtained in the year 2020, which has included …" → The suggested change will be considered, but we need to check that it doesn't change the idea.

L384: -> "water quality parameters" → "water' will be added

L385: Should wind speed also be considered? This parameter links to the oxygenation of shallow waters. → summer is not the period during which winds are the highest. In addition, high particulate load

limits exchange. *Abril G., Commarieu M.V., Sottolichio A., Bretel P., Guerin F. (2009) Turbidity limits gas exchange in a large macrotidal estuary, Estuarine, Coastal and Shelf Science, 83, 342-348.*

L381-389: Please provide a bit more information on how the path-analysis was conducted. How do we interpret the numbers shown on the arrows and the circles in Figure 8? Does the thickness of the arrows represent the strength of the correlation between the two variables? If so, please make it clearer in the caption. → Details on the patch analysis will be provided, in the method section and the figure caption will precise that the thickness of the arrows corresponds to the strength of the correlation. The correlations between DO and salinity/ discharge will be added, they were not plooted to avoid

L408: "less the attention" -> "less attention" …→ it will be corrected

L415-417: Do you mean the existence of turbidity maximum zone promotes severe deoxygenation? Why is temperature mentioned here? → Yes in macrotidal estuaries; hypoxia is not related to eutrophication but to warm and highly turbid waters.

L418-420: Got confused here. Why can low summer flows (with water stored in reservoirs) limit water warming? → water is stored in winter and released during the low summer flow to sustain river discharge. Sustaining summer river help to reduce the residence time of the water and to a less extend temperature rise (more stagnant water tends to warm up more when there is a lot of sunshine). And how can a lower summer flow increase the inflow of more oxygenated river water? As shown by the Saint-Savinien, the fluvial waters are more oxygenated, then the water released from fluvial reservoirs
Is also expected to be well oxygenated. Such scenarios have been modelled for the hyper-turbid Gironde estuary (Lajaunie-Salla et al Comparing the efficiency of hypoxia mitigation strategies in an urban, turbid tidal river via a coupled hydro-sedimentary–biogeochemical model Nat. Hazards Earth Syst. Sci., 19, 2551–2564, 2019 https://doi.org/10.5194/nhess-19-2551-2019. This section is be written more precisely to explain the low-water replenishment.

---

## Author Response (AR1)

We carefully considered the comments to produce an improved version of the manuscript. The main changes to the article are :

- The characteristics of macrotidal estuaries and the zone of maximum turbidity are more detailed.
- As the article focuses on dissolved oxygen, this parameter was not included in the first version, as we considered that detailing suspended matter concentrations was not an issue in a macrotidal estuary, where the presence of the TMZ implies very high SPM concentrations (> 1 g/L). Furthermore, this parameter was not measured in 2018 and 2019 as only DO and conductivity sensors were used. However, the comments of the two reviewers made us realise that very high SPMs are not so intuitive for colleagues not working in this type of environment. The SPM data have been added to Figures 2, 3 and 4.
- In addition to the changes made in response to the reviewers' comments, the manuscript has been carefully revised to improve the text.

The responses to the two reviewers are detailed below.
* * *
In black : comments of RC1 / in blue reply of authors

when the answer is based on the text of the article, the lines corresponding to the quotation are indicated as follows: Lines xx and the text is underlined.

The manuscript is like many others reporting low oxygen conditions where not measured before. The results are not particularly new given that deoxygenation occurs more and more, either because of poor water quality (usually eutrophication) or more collection of data.
We agree that the development of hypoxia is not a new idea.
We disagree with RC1 assertion that
- *results are not particularly new* : the dataset is new: the data has never been published before.
- *deoxygenation occurs more and more* : It's true that hypoxia is increasing in coastal areas, as reported in review articles. But RC1 seems to overlook the situation in estuaries, where restoration has improved dissolved oxygen levels after situations of hypoxia or even anoxia,

The idea of a turbidity maximum is presented as an issue, but there are few data that link the turbidity maximum and low oxygen The purpose of the article is not to discuss the maximum turbidity zone (TMZ) in the Charente estuary, for which references are given. Suspended particulate matter is now plotted in Figures 2, 3, and 4. There are already articles on the link between deoxygenation and TMZ in macrotidal estuaries (some references are given in the introduction).
Otherwise, the usual water quality parameters are those important in the development of the low oxygen—warmer waters, sluggish circulation, organic matter decomposition, especially from dense algal blooms. Lines 70-71 : SPM limits light penetration and hence photosynthetic activity. There is no dense algal blooms in the Charente estuary, you can see that particle concentration in TMZ is of several grammes of dry matter per litre.

There are poor language choices, poor logic statements; several examples were pointed out. These poor choices of words generate many illogical statements and poor understanding of the important issues and their positioning in the eventual interpretation of the results. There needs to be a thorough revision of the entire manuscript to use proper English, punctuation: before publication in biogeosciences discussion, the manuscript is reviewed by the editor in charge. If necessary, the editor requests corrections. There have been no requests for English corrections. However, if there are any typographical errors, we apologise and will correct them in the revised version.
consistency in references cited in the text and in the list of references. There are 27 references in total, and one is missing. This is the article by Curran and Henderson (1988), which is cited on line 347. The complete reference is: Curran, J.C., and Henderdon A.R.: The oxygen requirements of a polluted estuary for the establishment of a migratory salmon, Salmo salar L., population, J. Fish Biol. 33, 63-69n 1988.

The various methods used were not necessarily the best for defining low oxygen conditions. The results are a mixture of many different studies, methods, frequency of measurements, and methods to measure duration of low oxygen. For example, probes were placed 0.5 m below the surface. This is not a reasonable method to detect low oxygen conditions through the water column. The resources of concern, migrating fish use different depths of the water column, and thus potentially disrupted at different levels. The aim of this study was to detect possible hypoxia in the Charente estuary, which is the Charente estuary is (lines 93-94) a small, shallow, macrotidal estuary with an average tidal range of 4.2 m, reaching up to 7 m during spring tides. We hadn't considered the implications of the high tidal range and shallow depth of the estuary. We thought this was implicit and will correct this in the revised version. During each tidal cycle, the water level changes by several metres, associated at mid-tide with strong currents that tend to mix the entire water column. At low tide, the water level is lower, which explains the sentence lines 137-139 in the case of macrotidal rivers such as the Charente, where the pontoons are not far from the shore, they tend to be grounded at low tide, especially during spring tides; this was the case at L'Houmée, Rochefort and Matrou. Few depth profiles have been established, showing only slight differences between surface and bottom.

The 'high' frequency data are gappy, which is acknowledged, but too many conclusions are expressed in general terms rather than sound evidence, especially based on near-surface water data. We have never written that high frequency data is inaccurate, if that's what RV1 means by gappy. On the contrary, the comparison of two dataset (low and high frequencies) shows line 277 that high-frequency monitoring is essential in macrotidal estuaries.

Now that the authors have used the data to make predictions into the future, the federal (or provincial) agencies should fund an adequate monitoring program for spring, summer, and early fall. Recommendations from this study could inform what the program would include. Strategically placed profiling sondes for dissolved oxygen, temperature, and salinity information through the water column in at least a portion of the estuary for a volume-placed inquiry, especially including the bottom (or near-bottom) water where dissolved oxygen conditions are assumed to be critically low. We thank the reviewer for finding the work useful in providing recommendations to help regional water authorities implement appropriate monitoring. This is effective from 2021: section 4 Concluding remarks, lines 405-407 contains this information. There is an instrumented site (Tonnay-Charente). The multiparameter probe (sondes ?) also includes a turbidity sensor ; the monitoring is annual. But, in a shallow macrotidal estuary such as the Charente, the challenge is to determine the extent of the hypoxic zone, which will require the installation of additional sites, at least during the summer, rather depth profiles.

Section 4. Concluding Remarks is a correct, concise and well-presented summary of the reasons for the study, the basic results, and the implications for the future. The bulk of the manuscript is not any of these modifiers. Given the general tone of the review, we can speculate on the meaning of the second sentence. But we don't understand the word "modifiers" (nor do any of the online translators).

Specific comments:
The comments from lines 30 to 43 concern the abstract,
L 30, "these" does not identify the antecedent. The two larger estuaries, or multiple ones (not yet identified) between them.
Lines 28-30: Multi-year, multi-site, high-frequency water quality surveys have shown that the Loire, and to a lesser extent the Gironde, suffer from summer hypoxia. These observations raised the question of the possible occurrence of hypoxia, particularly in one of these …. → The demonstrative determiner "these observations" refers to surveys.

L 31, the first oyster producing area [by aquaculture?] we're talking about production area, so yes, aquaculture. Lines 101-103: The Charente river discharges through the Charente estuary (Figure 1) into the Bay of Marennes-Oléron, the first oyster-producing area in France (Goulletquer and Heral, 1997) and a major nursery ground for juveniles of the Bay of Biscay sole population (Le Pape et al., 2003; Modéran et al., 2012). Producing was replaced by farming to clarify.

L 31-33, assumes that the two other estuaries are compromised, but not yet established. We don't understand the comment. In the abstract it says lines 28-29 Multi-year, multi-site, high-frequency water quality surveys have shown that the Loire, and to a lesser extent the Gironde, suffer from summer hypoxia. Hypoxia events have already been measured in the other two estuaries: it is a fact. It is briefly developed and supported by references Lines 76-79 The spatio-temporal DO dynamics of the two large

systems are well described by long-term, high-frequency and multi-sites measurements of water quality. While episodic summer hypoxia events have been recorded in the fluvial Gironde estuary, the Loire estuary experiences permanent summer hypoxia in its lower reaches (Lanoux et al., 2013; Schmidt et al., 2019).

L 41-42, the fisheries comment seems peripheral, without any other substantiating information, but rather a mention of oyster production. Perhaps a broader reference to 'aquatic fishery resources.' In the Charente estuary, there is a real problem for migratory fish, especially during the downstream migration of larvae.

L 43, …Essential for the survival of aquatic animals and plants, oxygen levels…  [plants generate oxygen and are not compromised by lower dissolved oxygen] We admit that the sentence is very synthetic, the allusion is to chemical oxygen demand for decomposition of aquatic plants. Plants produce but also consume dissolved oxygen. Anyway, plants could be suppressed. "animal and plants" was replaced by "species".

L 50, also from advection of adjacent higher oxygen waters. Advection could be an input, or not if the advected waters are depleted in DO, but it is not a production of DO. Dissolved oxygen comes from two natural processes: diffusion from the atmosphere and photosynthesis by aquatic plants.

L 52, suggest "other aerobic organisms" instead of "decomposing organisms" Are the organisms doing the decomposing, or are they decomposing themselves through organic respiration. The purpose is was to describe briefly the DO budget, and not to detail the processes, Line 52 Conversely, bacteria and other decomposing organisms consume oxygen to break down organic matter. To stick to the broad general outlines, "and other decomposing organisms" was suppressed

L 54, oxygen saturation may also be used to determine autotrophy, respiration, and overall net production…Interesting but this is not but that's not the point of the sentence, which is simply an explanation of the term depletion: Dissolved oxygen depletion occurs when the measured dissolved oxygen concentration is below the theoretical saturation value

L 59 …long-term evolution… Geologic, millennial, decades, post-industrial period? In time with increased reactive N in the ecosystem? Generally speaking, for environmental issues, we're talking about the Anthropocene, when human activities have had a major impact on the environment since the late 18th century, as defined by Crutzen 2007. The timescales considered in this paper are the next few decades. its long-term evolution was replaced by "its evolution over several decades".

L 61, liquid and solid inputs from… Suggest "dissolved and particulate fluxes from…" the choice of words is a matter of discipline, the terms "liquid and solid inputs" are also used in estuaries and rivers

L63-55, long sentence with multiple ideas and no concise point The sentence lines 63-65 was shortened

L 68-70 is not clear. This reviewer follows the meaning, but it is not well presented. It's a long sentence to describe tidal pumping, the use of which would have raised even more questions for non-expert of TMZ. This was replaced by ". In macrotidal estuaries where the tidal range is greater than 3 m, tidal asymmetry (longer and weaker ebb currents; shorter and stronger flood currents) causes net upstream transport and trapping of suspended sediment in the inner and upper reaches (Allen et al., 1980). This results in the occurrence of a mobile region of high suspended particulate matter (SPM) concentration, known as the turbidity maximum zone (TMZ). The TMZ is characterised by SPM loads of several hundred milligrams to several grams of dry sediment per litre (Uncles et al., 2002). Such high particle concentrations affect water chemistry by limiting gas exchange with the atmosphere (Abril et al., 2009). In addition, high particle concentrations limit light penetration and hence phytoplankton primary production, but also provide substrate for microbial activity, promoting mineralisation of organic matter and DO consumption (Goosen et al., 1999). All this is likely to promote deoxygenation in macrotidal turbid estuaries (Talke et al., 2009)."

L 70-74 recognizes the difference between organic and inorganic suspended matter. Both are contained in the term SPM. The mineral turbidity or what is often considered the suspended sediments can be confounded by organic particulates in the form of phytoplankton, living and senescent, that may affect 'turbidity. we only briefly describe the well-know feature of turbidity maximum zone, and indeed several

grammes per litre of dry sediments (whatever the C content) SPM limits light penetration and hence photosynthetic activity, and gas exchange with the atmosphere. This is not the purpose to go in details of particle nature.

L 82, south-west in this sentence, versus north-east earlier in the manuscript. Are these geographic descriptors relevant to Bay of Biscay or the eastern coast of France?
Line 26: The French coast facing the Bay of Biscay (north-east Atlantic) is characterised by the presence of small macrotidal and turbid estuaries → The Bay of Biscay is located in the north-eastern part of the Atlantic Ocean
Line 80 for the south-west France: the Charente estuary is located in the south-west France and flows in the Bay of Biscay
The eastern coast of France is the continental Europe

L 94-95, suggest …. The tidal influence reaches 50 km upstream of the mouth but not further because of a dam at Saint-Savinien, which is opened during spring tides …. The proposed sentence does not really reproduce the idea of the initial sentence. We did not change the sentence.

L 99, is the word "longitudinally" necessary? The meaning is obscure, here and elsewhere. A trajectory in an estuary or river can be longitudinal (along the axis) or radial (perpendicular to the axis). We have used this term in previous articles without causing any problems for the editor or reviewers. For example, see the published article cited in this manuscrit : https://doi.org/10.3389/fmars.2019.00352

L 102, "Bay of Marennes-Oléron" should be identified in Fig. 1. Any geographic location in the text should be identified in Fig. 1. Bay of Marennes-Oléron was added to Figure 1

L 104, suggest…..The coastal fringe is more populated (80 to 100 inhabitants km$^{-2}$) than the rural interior (40 to 60 inhabitants km$^{-2}$). The largest…. Here again it is not exactly what we want to say, at least "densely" is missing in the suggestion.

L 108, words "low," "good" for water quality, "all uses are satisfied on average…" need to be replaced with more appropriate descriptors. Descriptors are defined, first line 46-48 Dissolved oxygen is one of the key physico-chemical quality elements that the EU Water Framework Directive (WFD) requires to be considered in order to achieve "**good** ecological status" (Best et al., 2007).
Again in the sentence Line 108 The target low flow, defined as "the reference flow which allows good water status to be achieved and above which all uses are satisfied on average 8 years out of 10", is 15 m3 s-1. Again, it refers to good status required by the EU. A statement on DO levels regarding such words is detailed section 2.2.4 Line 170 to 175.

AT THIS POINT, I WILL NOT MAKE ANY MORE WORDING SUGGESTIONS, UNLESS I DO NOT UNDERSTAND THE MEANING.

L 150+ future recommendation, if a probe is available with sufficient cable, a vertical profile would be advantageous compared to 0.5 m below the surface for defining low oxygen conditions. Interesting recommendation but technically and financially complicated to implement in a macrotidal estuary. In addition, at low tide the depth of the water column at most sites is less than 1-2 meter

L 159, define concentration below which the water is considered "hypoxic." From Fig. 1 appears to be 2 mg/l, which usually equates to 30% saturation, depending on the temperature and salinity. Figure 1 is a map, are we to understand that RC1 is referring to Fig. 2. We specify that the same colour is used for the figure 3 and 4 for the limit of 2 mg/L. It is now specified in the figure caption. But what a surprising question from a colleague who agreed to review an article on hypoxia. We use the commonly used threshold for hypoxia, ie a oxygen concentrations ≤2 mg/L (=63 µmol /L or), see for example Diaz and Rosenberg (2008). It is in fact written in the text line 56 even to hypoxia (< 2 mg L-1) or anoxia (0 mg L-1), or suggested line 248 The minimum DO reached at the beginning of August 2018 is 2.5 mg L-1, just above the hypoxic level.

L 160-165, long sentences could be shortened and more concise. The sentence was shortened.

L 194, is it is difficult to understand a 82% saturation coupled with a 2.2 mg/L dissolved oxygen concentration. This is of course a typing error for which we would apologize, but it will be corrected. It is 8.2%
L 200, are the oxygen values representative of 0.5 m below the surface? Yes, it is now precised in the caption of figure 2

I did not comment on the remainder of the manuscript but did read it. The comments would have continued as before. Just the main results; more concise; too many unnecessary details, unquantifiable terms. E.g., "rather similar" used twice to describe types of data, yes indeed Line 220 The seasonal pattern of DO variability is rather similar between the stations How would the reviewer qualify the differences, using statistics would unnecessarily complicate the discussion for data that are actually very similar? Rather was suppressed
 "rather dated." Line 365 is clear that the number of studies is limited and rather dated : The term "dated" does not apply to the data, but is used to describe the references cited in the review article d'Arevalo et al (2023), pointed out the need of studies.

The text was be carefully checked to make it more concise where necessary.
* * *
In black : comments of RC2 / in blue reply of authors

I read the comments from Reviewer 1 and responses from the authors. I agree with the authors that this article provides a new and valuable dataset and raises attention to the deoxygenation situation of macrotidal estuaries. I also agree with Reviewer 1 that authors can further improve the clarity of statements and word choices (detailed in my specific comments) and include necessary information to help readers interpret results (e.g., SPM data to show the maximum turbidity zone, water depths of monitoring stations to justify using surface data alone). Reading the comments made by the two assessors made us realise that it was necessary to add information about the sites, macrotidal estuaries and the zone of maximum turbidity. This will be corrected in the revised version. We would like to thank the reviewer 2 for their careful reading and the suggestions and questions that followed. They will greatly improve the manuscript.

The turbidity maximum zone (TMZ) appeared on many occasions throughout the Results and Discussion section. However, there is no data or analysis on TMZ presented in the manuscript. According to Sections 3.1 and 3.5, the concentration of suspended particulate matter (SPM) is monitored at several stations and used to conduct path-analysis. However, the data of SPM is not shown anywhere. I suggest presenting the SPM data to directly support the analysis of TMZ and the link between SPM and oxygen dynamics. The reviewer is right, there are indeed measurements of the suspended matter load that were used for the analysis. The low-frequency monitoring carried out by the Water Agency includes this parameter. As the article focuses on dissolved oxygen, we did not include this parameter. Figures 2, 3 and 4 have been modified to include the suspended matter load.

The manuscript mainly presents and discusses variations of temperature, salinity, and dissolved oxygen (DO) in Charente. It suggests that temperature is the main controlling factor of DO. What are the conditions of other water quality parameters, such as inorganic nutrients, organic matter, and chlorophyll, in this estuary? Do they contribute to the low oxygen events in the Charente?
The analysis also suggests that the downstream estuarine water has a lower oxygen level while the upstream river water has a higher oxygen level. This spatial variability doesn't seem to be controlled by temperature. What are the other driving factors? In fact, few studies have been carried out in the Charente estuary, so there is little data available. This initial work was initiated following the occurrence of hypoxia in neighbouring macro-tidal estuaries. At this stage, the work has revealed summer hypoxia. The link with temperature is strong, and in autumn, although SPM is high, as soon as the temperature drops, oxygenation is restored to levels above 5 mg/L. In the upstream estuary, measurements were done at Saint-Savinien, which is very closed to the extension of the estuary and then this site is influenced by riverine waters and is not impacted by the TMZ presence. For further interpretation, more detailed studies of particles (% carbon, lability) will be required. As far as chlorophyll is concerned, SPMs of several grams per litre exclude primary production.

In addition, Figure 4 shows that the measured oxygen level is way below the saturated oxygen level during summer. This suggests remarkable oxygen depletion, which can be related to the biological and chemical processes consuming oxygen. Are they due to microbial degradation of organic matter? While the oxygen-consuming processes are also affected by temperature, there should be sources of organic matter to fuel the microbial degradation. Therefore, I think other parameters in addition to temperature and salinity are also required to better understand the variations of DO. Indeed, in TMZ oxygen consumption is mainly related to microbial degradation, a process that is greatly influenced by temperature. Even if the carbon load is low, the huge charge of particles (several grams per litre) implies an important oxygen consumption. The discharge was considered in the path-analysis as it influences directly water renewal, but its impact was low regarding the other considered parameters. The impact of discharge on dissolved oxygen will be added in Figure 8. The same feature is observed in the other nearby macrotidal estuaries: the Loire in the north (Schmidt et al, 2020) and in the south the Gironde-Garonne system. In the Gironde-Garonne system, a coupled hydro sedimentary-biogeochemical model of DO concluded that "simulates an intensification of summer hypoxia with an increase in temperature, a decrease in river flow or an increase in the local population, but not with sea level rise, which has a negligible impact on dissolved oxygen. Different scenarios were tested by combining these different factors according to the regional projections for 2050 and 2100. All the simulations showed a trend toward a spatial and temporal extension of summer hypoxia due to temperature rise and discharge decrease. An increase in temperature accelerates oxygen consumption by biogeochemical reactions.

Specific comments
L31: the first oyster-farming area? → done

L35: "salinity sensors"? Do you mean conductivity sensors? → conductivity added, the HOBO Salt Water Conductivity/Salinity Data Loggers measure conductivity, and temperature, salinity is calculated https://www.onsetcomp.com/products/data-loggers/u24-002-c And what are used to measure temperature? HOBO Dissolved Oxygen Data Loggers measured dissolved oxygen and temperature.

L36: "will be" -> "is" → Done

L36: specify the sampling frequency of the "low-frequency" dataset → "8-12 measurements per year" added. The information was already given in the section 2.2.3 in the first version.

L37: "a degradation of oxygenation" -> "a deoxygenation trend" It is not exactly the idea, "degradation" was replaced by "worsening"

L41: -> "main controlling factor of DO" → Done

L46: "diagnosing of" -> "diagnosing" → corrected, "of" was suppressed according to for diagnosing  water quality

L60: "less the attention" -> "less attention" → corrected

L60: Not sure how we can conclude that "hypoxia in estuaries has received less the attention than in coastal waters". → just a rapid check: 775 results from Web of Science Core Collection for estuary/hypoxia (considering that some of the articles concern more the outer estuary (= coastal water) and not inner estuary). Only 14 references in Gilbert and al 2010.
922 + 174 for coastal water and continental shelf/hypoxia, Anyway, the sentence was modified as the trajectories of estuaries is in fact more complicate than in coastal waters. This aspect is now detailed and the sentence is replaced by '*In particular, predicting DO trajectories in estuaries is more complex than in coastal waters due to hydroclimatic but also anthropogenic forcing (Villate et al., 2013). Indeed, during the 20th century, many estuaries adjacent to cities have experienced significant water quality degradation due to excessive discharges of untreated domestic and industrial wastewater. Severe hypoxia was observed in estuaries such as the lower Hudson, Thames and Scheldt, among others (Brosnan and O'Shea, 1996; Billen et al., 2005). In the 1980s and 1990s, improvements in the collection and treatment of wastewater helped to improve DO levels in European estuaries (Andrews and Rickard, 1980; García-Barcina et al., 2006).*'

L67: "given their large impact on aquatic organisms" – can be deleted. It has been elaborated by the preceding sentences. → suppressed

L81-84: please provide references for these predicted changes. → relevant references added; "and the population growth forecast for the south-west France by 2030" was suppressed in the absence of publications in English. There is only French reports on the topic. Anyway the global change already will effect DO.

L131: the depth of the selected sites should be described to help readers better interpret the data collected at 50 cm below the surface. Maybe mark them in Figure 1. → range of water depth was added in table 1

L135: What parameters are measured through the multi-parameter sensor? → temperature, conductivity/salinity, turbidity, dissolved oxygen, this information was added.

L154: consistently use "50 cm" or "0.5 m" → "0.5 m" is used in the whole text

L160: "Charente estuary is not known to have hypoxic situation" - is this statement based on monitoring or the geographic, physical, and/or chemical characteristics of the estuary that make it not prone to develop hypoxia? It is based on the absence of publication. The sentence was clarified "whereas no publication reported such a problem for the Charente estuary, unlike the Loire and Gironde estuaries".

L161-170: What methods are used to collect these low-frequency data? And at what depths are the measurements conducted? Are they comparable to those of the high-frequency data? These data are produced by certified laboratories that produced qualified data. Measurements are also made in surface waters. Information added in the text.

L175: delete "that" → Done

181: "varies" -> "vary" → discharges → discharge

187-188: This should be divided into two sentences. → the sentence was modified as SPM is now plotted in Figure 2 and 3.

L197: insert "with" before "DO reaching …"→ Done

L200: the unit of saturation in Fig. 2(d) is incorrect. The time period of the observations at Saint-Savinien is missing. …"→ Corrected

L217: delete "even" …"→ the sentence was modified to better convey the idea: "The minimum DO recorded in Tonnay-Charente is characterised by the following (Figure 2d): while the lowest value (3.8 mg L-1) was observed on 27 July 2012 during the period 2011-2015, there is a trend towards concentrations below 3.5 mg $L^{-1}$ almost every year since 2016."

L230: Fig. 4a. The shaded area of discharge is covered by that of salinity from time to time. Suggest plotting the discharge using a solid line on the top of salinity. …"→ the figure 4a was modified as suggested
Fig. 4d. The legend of the gray line could be modified as "DO at saturation". "DO-100%" can be misinterpreted as the "DO saturation – 100%" with a unit in percentage. …"→ it is written in the figure caption, but the legend was changed as 'theorical DO at saturation'

L240: In Figure 4, where can we tell "the presence of a moderate TMZ"? → the presence of the TMZ is related to the fluvial discharge. SPM was added in Figure 4. Moderate TMZ is about 2-3 g/L, considering that it could be more more than 6-8 g/L. SPM is now described as Figures 2, 3 and 4 show also SPM.

L248: considering -> consider …→ Corrected

L251, 257: will be -> was → Corrected

L250-260: Please pay attention to the tense being used. It should be consistent throughout the manuscript. …→ it was checked
L260: Please include the unit. → Done

L262: "..allowed TMZ to develop and persist until the autumn floods (Fig. 4)." – this cannot be inferred from Figure 4. → SPM is now plotted in Figure 4

L270: "This validates the use of these data …" -> "The agreement between the two datasets validates the use of the low-frequency data …"→ Corrected

L273: "inability of the low-frequency data to …"→ the low-frequency data → Added

L282: What trends of what variables? Figure 5 only shows dissolved oxygen. L295: Figure 5 – suggest marking the period of high and low tides on the time series. …"→ Figure 5 was modified and a second figure with zoom on tidal cycle was added

L302: The previous paragraph (L284-286) suggests that at high tide, the advection of oxygen-poor water from downstream leads to low-oxygen condition. But this sentence suggests the opposite. L305: This contradicts L284-290 which suggests that the downstream estuarine water has a lower DO concentration while the upstream river water has a higher DO concentration → 1it was clarified and we hope that the modified figure 5 and the new figure 6 will help

L310: "Oxygen" -> "oxygen" → Corrected Do you mean "localization" or "location" of the oxygen minimum zone? -> Location

L318-319: Do you mean the DO data collected in the summer of the 3 years shows a decreasing trend? No, it is written "In Tonnay-Charente, the 2018-2020 high-frequency monitoring confirms the decreasing trend in summer DO detected by the low-frequency monitoring': no the idea is to indicate that high-frequency data confirms that DO concentration are lower in summer. The sentence was reformulated to clarify;
The time series presented in Figure 4 is not obvious to support this statement. The data in 2018 only covers 3 weeks, which is not sufficient to represent the entire summer condition of 2018. → no, we want to say that DO decrease in summer as observed by low-frequency monitoring. The sentences will be clarified "In Tonnay-Charente, high-frequency monitoring in 2018-2020 confirms the occurrence of low oxygen levels in summer, as shown by low-frequency monitoring since 2011."

L336-337: We need the information on the depths of these monitoring sites. Are there any data to show how stratified the water column is? …"→ Information on the depth of the sites have been added: typically, the water column is between 1-2 at low tide to 8-10 at high tide. Considering the high tide currents, there is no or limited stratification in macrotidal estuaries, especially in the upper section. In addition, the sentence was just to remind that measurements are done in surface water, and not in depth, suggesting that DO at the bottom could be even lower. Then a stratification would isolate even more the bottom waters, this is no issue then regarding stratification.

L346: "reach borderline or even hypoxic, levels" -> "reach borderline or even hypoxic level" …"→ ',' was suppressed L353: delete "comprised"→ Delete L355: delete the second "target" → Delete
L356: -> "summer levels of DO": → DO added

L365: "However, it is clear that the number of studies is limited and rather dated. " – not clear to me how we can come to this conclusion. The cited references in the preceding sentences are mostly between 2021 and 2023. → Yes, indeed and the most recent is Arevalo et al. (2023) that presents a review of the literature focusing on temperate estuaries. Therefore, the recent references do not rule out the fact that a large part of the works is not of recent date.

L382: "used the 2020 dataset, which has the interest to include also …" -> "used the dataset obtained in the year 2020, which has included …" → The text has been replaced by "For a first estimation of the controlling factor of the DO variability using a path-analysis, we used only high frequency data set from 2020. This has the advantage of including SPM concentrations in addition to temperature, salinity and DO data".

L384: -> "water quality parameters" → "water' added

L385: Should wind speed also be considered? This parameter links to the oxygenation of shallow waters. → summer is not the period during which winds are the highest. In addition, high particulate load limits gas exchange. *Abril G., Commarieu M.V., Sottolichio A., Bretel P., Guerin F. (2009) Turbidity limits gas exchange in a large macrotidal estuary, Estuarine, Coastal and Shelf Science, 83, 342-348.* Later, more comprehensive studies, particularly if multi-year modelling of DO is undertaken, could take this parameter into account. Anyway the figure 8 shows that the considered parameters explain 96% of DO variaibility.

L381-389: Please provide a bit more information on how the path-analysis was conducted. → Details on the patch analysis were added in a new sub section 2.2.5 in the method  How do we interpret the numbers shown on the arrows and the circles in Figure 8? The numbers correspond to the regression coefficient and the numbers in the circles correspond to the % of variability explained by the parameters taken into account. Does the thickness of the arrows represent the strength of the correlation between the two variables? If so, please make it clearer in the caption. Yes, the figure caption precises now that the thickness of the arrows corresponds to the strength of the correlation. The path analysis has been modified to show the relationship between all the considered parameters. The fact that tidal range and discharge have little influence was only written. Initially, the idea was to present a simplified schema to show the interaction discharge/ tidal range/spm an spm/temperature/salinity/DO. Now all the correlations are plotted with two different designs (line and dotted line). It is explained in the caption.

L408: "less the attention" -> "less attention" …→ The sentence was replaced by 'Reduction in wastewater pollution in recent decades have helped to improve oxygen levels in estuaries. However, the trajectory of estuarine DO is difficult to predict and will depend on their vulnerability to the effects of climate change'

L415-417: Do you mean the existence of turbidity maximum zone promotes severe deoxygenation? Why is temperature mentioned here? → Yes, in macrotidal estuaries; hypoxia is not related to eutrophication but to warm and highly turbid waters. The main forcing factor is temperature, amplified by TMZ.

L418-420: Got confused here. Why can low summer flows (with water stored in reservoirs) limit water warming? → water is stored in winter and released during the low summer flow to sustain river discharge. Even if the impact would be limited, sustaining summer river help to reduce the residence time of the water and to a less extend temperature rise (more stagnant water tends to warm up more when there is a lot of sunshine). However this was suppressed And how can a lower summer flow increase the inflow of more oxygenated river water? As shown by the Saint-Savinien, the fluvial waters are more oxygenated, then the water released from fluvial reservoirs also expected to be well oxygenated. Such scenarios have been modelled for the hyper-turbid Gironde estuary (Lajaunie-Salla et al Comparing the efficiency of hypoxia mitigation strategies in an urban, turbid tidal river via a coupled hydro-sedimentary–biogeochemical model Nat. Hazards Earth Syst. Sci., 19, 2551–2564, 2019 https://doi.org/10.5194/nhess-19-2551-2019. This section is be written more precisely to explain the low-wat

---

## Author Response (AR2)

Please find the replies to the questions/concerns of the reviewer. We have also modified colors of figures 5 and 6

This manuscript is generally well-written and interesting and gives a lot of information on the composition and state of estuaries in this region. The figures are good and insightful. I have a few outstanding questions/concerns, listed below. Once those have been addressed, I think this manuscript will be ready for publication. → Thanks for this evaluation.

One overarching comment I have has to do with vertical properties of the water column. All measurements are taken near the surface. Is there evidence to show that surface properties represent those throughout the water column? In other words, does hypoxia at 0.5 m also mean that there is hypoxia at the bottom? → This question was already addressed in the first review round. Additional infomations have been already added such as the fact that *the Charente estuary is a macrotidal estuary with an average depth of approximately 9 m around Rochefort. It is a partially mixed to well-mixed macrotidal estuary, with stratified conditions occurring at very high river discharge (Toublanc et al., 2016). The tides are semidiurnal with an average amplitude of 4.2 m, reaching up to 7 m during spring tides.* Information on the depth of the sites have been added: typically, the water column is between 1-2 at low tide to 8-10 at high tide. Considering the high tide currents and the limited depth, especiallay at low tide, there is no or limited stratification in such macrotidal estuary, especially in the upper section. Measurements are done in surface water, and not in depth, then one could consider that the surface values could be rather representative of DO. However at the bottom DO could be even lower (as bottom waters are isolated from the atmosphere and the occurrence of high SPM (several g / L) not only precludes primary production but also promotes consumption. A stratification would isolate even more the bottom waters. During some field works, we did vertical profiles that show deep DO to be always lower than surface DO.

- Line 35 – should temperature be included here? → Temperature was added although stricto sensus there were no specific temperature sensor, it is the optode (DO sensor) that measures DO and temperature.

- Lines 49 to 53 – I found this confusing to read. I suggest focusing it to discuss processes that add oxygen and processes that remove oxygen → I don't understand the comment as the lines from 49 to 51 detail processes that add DO (« or adsorbed » was suppresed to simplify the sentences). Then the lines 51 to 52 explains processes removing DO.

- Line 104 – What is a turbidity maximum zone (TMZ) and how is it defined? → The notion of TMZ is previously defined in the introduction from lines 70 to 78.

- Figure 1 – I suggest adding the catchment area to this figure → the water cathment area is now indicated, and also the extent of the study area and of the oxygen minimum zone.

- Lines 127 to 131 – The authors mention that there are 3 datasets but them didn't really specify the 3. Please clarify. → The 3 datasets are described in each sub-sections :

2.2.1 High-frequency summer monitoring of the Charente estuary
2.2.2. Longitudinal investigation of the Charente estuary
2.2.3 Low-frequency long-term monitoring
The details were already given in Table 1, « There are three datasets» is now added to the table caption

- Line 137 – Should temperature be included here? → Optode measures always dissolved oxygen + temperature whatever the supplier (HOBO, RBR, NKE..).

Also, what is the accuracy of the HOBO optode and conductivity sensor? And how does the error estimate of these sensors compare with the variability of the data?

The specification of the Hobo sensors are given on the Hobo web site https://www.onsetcomp.com/products/data-loggers/u26-001#specifications:

DO measurement range 0 to 30 mg/L ± 0.2 mg/L up to 8 mg/L; ± 0.5 mg/L from 8 to 20 mg/L

Temperature Measurement: -5 to 40°C ± 0.2°C

The range of values recorded in this work (LS and HF) is 1 to 10 mg L$^{-1}$, the sensor errors (0.2 at values < 8 mg L$^{-1}$) are negligible considering the range of variation.

The conductivity-salinity sensor has a measurement range from nearly 0 to 40, with a precision of 3%. The salinity range of the data acquired during the high-frequency and longitudinal survey is 1 to about 15), there is no problem of sensitivity of the conductivity /salinity sensor, even during a tidal cycle. It has to be said that conductivity/salinity (along with temperature) is one of the most reliably measured physical parameters, even at low levels of variation, which is not the case here.

A sentence is added

*The measured parameters are temperature (−-5 to +35°C; ± < 1%), conductivity/salinity (salinity range 0.1 -42; ± < 5%), turbidity (Turner Designs CYCLOPS-7; 0 - 3000 NTU; ± < 5%) and dissolved oxygen (0 - 20 mg L$^{-1}$; ±0.1 mg L$^{-1}$)*

- Lines 143 to 145 – Similar to above, what is the accuracy of the SAMBAT instrument? And how does it compare to the HOBO? → These informations are already provided along with the dataset at the link doi.org/10.17882/95886 provided in table 1. The sentence below is now added in the method section:

*The measured parameters are temperature (−-5 to +35°C; ± < 1%), conductivity/salinity (salinity range 0.1 -42; ± < 5%), turbidity (Turner Designs CYCLOPS-7; 0 - 3000 NTU; ± < 5%) and dissolved oxygen (0 - 20 mg L$^{-1}$; ±0.1 mg L$^{-1}$)*

*The Sambat accuracy is even better than those of Hobo sensors.*

[Figure]

- Lines 145 to 146 – The conversion of turbidity from voltage to g/L is not trivial. Could the authors please show how this did this? Figures would be great so the reader can assess the accuracy of the

conversion. → The method did not indicate the the turbidity was in voltage, but in NTU. The specification of NKE Instrument is 0 - 3000 NTU. The sensor is selled with a calibration using formazine (3 points 10 – 100 – 1000 NTU). Above 1000 NTU the relationship between NTU and formazine (FNU) is no more linear. Anyway, turbidity (FNU or NTU) is not very informative. It is the reason for which a relationship is establised between NTU and SPM. It is done at the lab using different concentrations of SPM from the Charente estuary (SPM determined by classical gravimetric method). Below an example of a SPM-to-turbidity rating curve obtained on estuarine SPM. Such determinations are repeated to produce a mean relationship to convert NTU in SPM.

- Line 157 – I haven't heard about a longitudinal study before. Could you please add a description of how this is done? Because I am not familiar with this method, I struggled to interpret the results. → we are confused by this question because it is a formulation that has already been used in previous articles without raising any questions (see for example https://www.frontiersin.org/articles/10.3389/fmars.2019.00352/full). The Collins dictionary defines longitudinal as "A longitudinal line or structure goes from one end of an object to the other rather than across it from side to side". Applied to a river or estuary, this indicates a transect from one site (e.g. Soubise) to another (e.g. Tonnay-Charente) along the axis of the estuary. This is in contrast to a radial (which would have been made perpendicular to a site, across the width of the estuary).

Section 2.2.2 relative to the longitudinal investigation was modified in order to simplify the text.

- Line 198 – What is TMZ? Could you please define it again here so the readers don't have to dig through the methods section → TMZ is turbidity maximum zone and this feature of macrotidal estuary is explained in the introduciton (and not the method !) . The sentence «The TMZ is present at Rochefort at least from July to November » is replaced by « The turbidity maximum zone (TMZ), corresponding to an SPM concentration greater than 1 g L$^{-1}$, is present at Rochefort at least from July to November ». The threshold of 1 g L$^{-1}$ is now indicated in the Figures 2, 3 and 4.

[Figure]

- Figure 4 – It is really interesting to me that temperature doesn't vary with tides yet temperature and salinity do. Do the authors have some explanation about why temperature doesn't vary with the tides? Or why there isn't day/night heating and cooling? → We are in a macrotidal estuary with a tidal range of over 3 metres and strong mid-tide currents. Line 100-101, it is indicated that "It is a partially mixed to well-mixed macrotidal estuary, with stratified conditions occurring at very high river discharge ". "The tides are semidiurnal" this high mixing produces a rather homogeneity on temperature. However, we are not agreeing the comment, temperature does not present as high variability than salinity (we are in an estuary), but there are short-term variabilities with tide. It is less obvious because the annual

range of temperature is large, however during a tide, the temperature difference could be of 1-2°C, even more during heat waves, for example in 2018 with an increase of more than 2°C (the figure is a zoom of 2018 and 209 of the figure 4

- Line 267 – Are the authors referring to Figure 4a here, not Figure 3? → indeed it is corrected

- Line 302 – I find this paragraph confusing. For example, I don't see Saint-Savinien data in Figures 5 or 6: → this refers to the previously described data at Saint-Savinien. "Fig.2d" is added and the sentences were modified to clarify the purpose.

- Line 313 – The authors state upstream but I think that they mean downstream (i.e. closer to the ocean)? → corrected

- Line 319 – What is the proof that the waters around Rochefort are always surrounded by oxygen-poor water? → the different dataset show that DO is also low at the two stations (Tonnay-Charente, Matrou) that surrounding this site (Rochefort) (Figure 5 and 6).

- Section 3.3 – I found this section very difficult to read, possibly because I didn't understand how longitudinal distributions were calculated. Other questions I had in this section are:

o How was the OMZ defined? How deep is it? → it is not OMZ as in open ocean, it is similar to the turbidity maxim zone TMZ, ie the estuarine region where DO is low. To avoid confusion, it is changed to eOMZ (estuarine OMZ).

o I struggled to picture where on a map the low oxygen waters were. I suggest that the authors add a map to Figure 7 that shows the location and magnitude of the low oxygen waters. → indication of the study area and of the OMZ is added in the map in Figure 1.

o Lines 340 to 345 – This is very confusing and I couldn't figure out exactly where the high and low oxygen waters were. This is partly because I think the upstream and downstream labels may be wrong (or different from the way I think – in my mind, downstream means closer to the ocean and upstream means further away from the ocean) and also because talking about low oxygen water at L'Houmee is different than what I see in Figure 5. → this was checked, it is ok. This is precsied that low DO at L'Houmée are observed at low tide. Names of sites were added.

- Line 368 – Where does this 25 km extent come from? Again, showing these data on a map would help the reader picture the low oxygen zones → The position and distance of stations is also given in figure 7. An arrow was added to help the reader.

- Figure 8b – This figure reminds me of Figure 3f from Rosen et al (https://www.frontiersin.org/articles/10.3389/fmars.2022.1000041/full) – can a comparison be made? → THe two figures don't correspond to the same information, in Rosen et al, it corresponds to the hypoxia% in the upper 50 m, whereas figure 8 shows the number of hours per day during which DO is in specific ranges. The only possible adaptation is to calculate the % of the day hypoxia occurs. For clarity we had a figure, it provides the same pattern as the figure with the number of hours. The difference is the unit , % instead hour. We are not convinced that it is useful, but the figure is changed if it could help reader to understand.

---

## Author Response (AR3)

**Public justification (visible to the public if the article is accepted and published)**:
Dear Dr. Schmidt and Dr. Diallo,

thank you for submitting your revised manuscript. I accept your manuscript for publication subject to only some minor technical corrections:

Thanks for the feedback, below the replies to the comment in blue

- Early in the manuscript you mention several times 'aquatic plants'. The term 'plants' would exclude photosynthetic prokaryotes such as cyanobacteria. I suggest a change to 'aquatic primary producer' or 'aquatic photosynthetic organisms'. → Done, «aquatic photosynthetic organisms» replaces plants in lines 50 and 54. The sentence beginning on line 56 has been amended to delete the term "plants".

- Figure 1 caption: Please delete "and the oxygen minimum zone". If the extent of your study did not find the disappearance of the OMZ then you are unable to say how far the OMZ extended. → Done. The corresponding arrow on the map has also been suppressed. A change was made as indicated in the figure caption: "The underlined city names correspond to the locations where hypoxia levels were measured during the study" was added

- Table 1 caption: Please define SPM → SPM has been replaced by suspended particulate matter. The same term is now used in line 150 to harmonise the name.

- Table 1: All htpps links need the date of last access → The date of the last access is added in the caption of Table 1

- Please introduce the term eOMZ in the introduction and use it consequently throughout the manuscript (there are several cases where you still use 'oxygen minimum zone'). → Done

- Appendix A: https needs the date of last access

- I further noticed an accessive use of 'This' at the beginning of sentences without defining what 'this' is referring to (i.e., a subject is missing). This method is difficult for the reading flow because it forces the reader to interpret what you are referring to. Please consider adding subjects. → the number of "this" was reduced, with occasional rewording